# Analog optical computer for AI inference and combinatorial optimization

Kirill P. Kalinin[1✉], Jannes Gladrow[1], Jiaqi Chu[1], James H. Clegg[1], Daniel Cletheroe[1], Douglas J. Kelly[1], Babak Rahmani[1], Grace Brennan[1], Burcu Canakci[1], Fabian Falck[1], Michael Hansen[2], Jim Kleewein[2], Heiner Kremer[1], Greg O'Shea[1], Lucinda Pickup[1], Saravan Rajmohan[2], Ant Rowstron[1], Victor Ruhle[2], Lee Braine[3], Shrirang Khedekar[3], Natalia G. Berloff[4], Christos Gkantsidis[1], Francesca Parmigiani[1✉] & Hitesh Ballani[1✉]

Artificial intelligence (AI) and combinatorial optimization drive applications across science and industry, but their increasing energy demands challenge the sustainability of digital computing. Most unconventional computing systems[1–7] target either AI or optimization workloads and rely on frequent, energy-intensive digital conversions, limiting efficiency. These systems also face application-hardware mismatches, whether handling memory-bottlenecked neural models, mapping real-world optimization problems or contending with inherent analog noise. Here we introduce an analog optical computer (AOC) that combines analog electronics and three-dimensional optics to accelerate AI inference and combinatorial optimization in a single platform. This dual-domain capability is enabled by a rapid fixed-point search, which avoids digital conversions and enhances noise robustness. With this fixed-point abstraction, the AOC implements emerging compute-bound neural models with recursive reasoning potential and realizes an advanced gradient-descent approach for expressive optimization. We demonstrate the benefits of co-designing the hardware and abstraction, echoing the co-evolution of digital accelerators and deep learning models, through four case studies: image classification, nonlinear regression, medical image reconstruction and financial transaction settlement. Built with scalable, consumer-grade technologies, the AOC paves a promising path for faster and sustainable computing. Its native support for iterative, compute-intensive models offers a scalable analog platform for fostering future innovation in AI and optimization.

Computing today is digital, but analog has a future. Exponential advances in digital hardware have both driven and benefited from the rise of artificial intelligence (AI), but its escalating energy and latency demands push digital specialization to its limits[8]. Analog approaches—leveraging optics[1–4,9], analog electronic crossbars[5,6] and quantum annealers[7]—promise orders-of-magnitude gains in efficiency and speed. Existing hardware demonstrations focus either on AI inference[1–3,10–13], which accounts for 90% of energy in commercial deployments[14], or combinatorial optimization[7,15], but none efficiently accelerate both on the same analog hardware.

Here we introduce the analog optical computer (AOC), a non-traditional computing platform designed for both AI inference and combinatorial optimization. By combining optical and analog electronic components within a feedback loop, the AOC rapidly performs a fixed-point search without digital conversions. In each loop iteration of approximately 20 ns, optics handle matrix–vector multiplications, whereas analog electronics perform nonlinear operations, subtraction and annealing (Fig. 1a–c). Over multiple iterations, the fixed-point nature of the AOC enhances noise robustness, which is essential for analog hardware.

The AOC's fully analog architecture and fixed-point abstraction address two key challenges in unconventional computing. First, hybrid architectures typically accelerate linear operations but rely on digital nonlinearities, resulting in energy-intensive conversions[2,13], which are eliminated in the AOC. Second, they often face an application-hardware gap: memory-bound AI models are hard to accelerate[14], and prevalent binary optimization formulations limit practical applicability[16]. With its unifying fixed-point abstraction, the AOC closes this gap (Fig. 1d,e).

For inference, the AOC accelerates emerging iterative models, including fixed-point models such as deep-equilibrium networks[17], which are compute-bound and costly on digital chips but naturally suited for the AOC. These models enable iterative reasoning with dynamic inference time computation[18,19]. For optimization, the AOC supports quadratic unconstrained mixed optimization (QUMO), a flexible formulation with binary and continuous variables that captures real-world problems[20].

[1]Microsoft Research, Cambridge, UK. [2]Microsoft, Redmond, WA, USA. [3]Chief Technology Office, Barclays, London, UK. [4]Department of Applied Mathematics and Theoretical Physics, University of Cambridge, Cambridge, UK. ✉e-mail: kkalinin@microsoft.com; francesca.parmigiani@microsoft.com; hitesh.ballani@microsoft.com

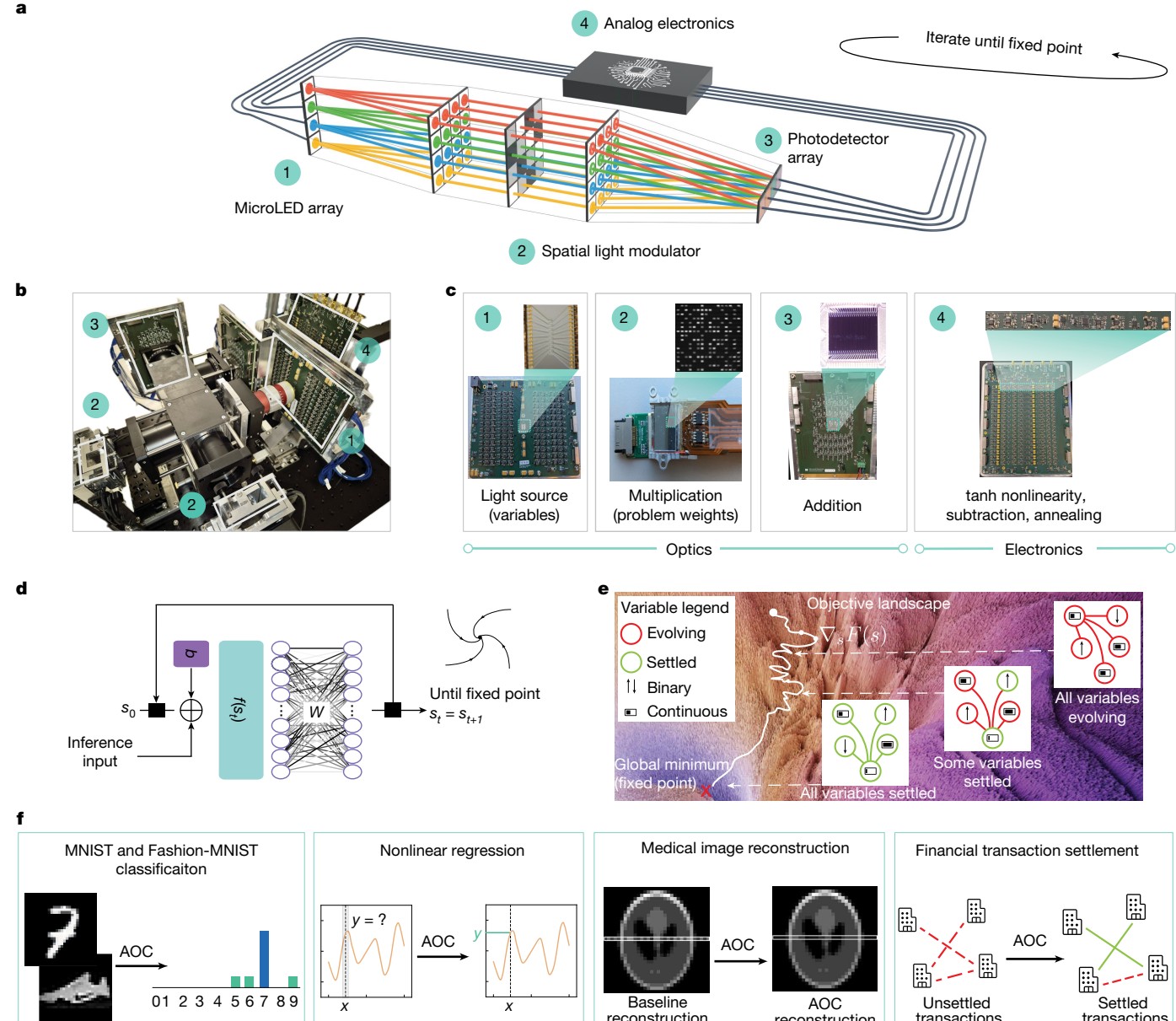

**Fig. 1 | The AOC and its applications. a**, A schematic of the AOC hardware architecture with key components 1–4. **b**, The AOC hardware architecture. **c**, Key hardware components. The microLED array (1) is the light source and represents neural network activations or optimization variables. The spatial light modulator (2) stores neural network weights or optimization problem coefficients, and multiplies them with the incoming light. The photodetector array (3) adds and transfers optical signals into the analog electronic domain. The nonlinearity, subtraction, annealing and other computations are applied in analog electronics (4). **d**, ML inference as a fixed-point search. The AOC hardware is used to accelerate ML inference, using an ML equilibrium model that entails finding their fixed points. **e**, Quadratic optimization as a fixed-point search. A schematic of the convergence to minima over time for an optimization problem with continuous (horizontal bars) and binary (vertical arrows) variables, experiencing gradients (red) from other variables until convergence to the fixed point (green). **f**, Applications realized on the AOC hardware. The AOC hardware performs inference for MNIST and Fashion-MNIST classification tasks and nonlinear regression, and also solves industrial optimization problems, including medical image reconstruction and transaction settlement between financial institutions.

The current small-scale AOC accelerates equilibrium models with up to 4,096 weights at 9-bit precision, performing image classification and nonlinear regression tasks (Fig. 1f). To enable these inference tasks, we design a differentiable digital twin (AOC-DT) that achieves over 99% correspondence with the physical hardware. For optimization, the AOC solves QUMO problems with up to 64 variables, tackling real-world applications such as medical image reconstruction and financial transaction settlement. The AOC-DT is used to demonstrate scalable solutions for industrial problems, such as reconstruction of a brain scan with over 200,000 problem variables. Compared with well-known heuristics[21] and commercial solvers[22], we set state-of-the-art results on several instances from a standard quadratic optimization benchmark[23].

Designed with consumer-grade optical and electronic components, the AOC leverages mature manufacturing processes, with future scalability relying on tighter coupling of integrated analog electronics with integrated three-dimensional (3D) optics. By eliminating digital–analog conversions and merging compute and memory to bypass the von Neumann bottleneck, the AOC can achieve substantial efficiency gains albeit specialized. With projected performance around 500 tera-operations per second (TOPS) per watt at 8-bit precision—over 100-times

more efficient than leading graphics processing units (GPUs)[24]—the AOC represents a promising step towards sustainable computing.

## Fixed-point abstraction

The AOC hardware unifies machine learning (ML) inference and optimization paradigms through an iterative fixed-point search. In ML, the inference of equilibrium[17] and energy-based[25] models entails finding their fixed points, whereas in optimization, objective minima represent fixed points of gradient-descent-based methods. The core AOC abstraction realizes the following iterative update rule:

$$\mathbf{s}_{t+1} = \alpha(t)\mathbf{s}_t + \beta W f(\mathbf{s}_t) + \gamma(\mathbf{s}_t - \mathbf{s}_{t-1}) + \mathbf{b}. \tag{1}$$

At each iteration $t$, the continuous real-valued state vector $\mathbf{s}_t \in \mathbb{R}^N$ is updated to $\mathbf{s}_{t+1}$, with each iteration corresponding to a signal round-trip time in hardware. Although equation (1) is discrete, the AOC operates in a continuous clock-free manner. The annealing schedule $\alpha(t): [0, T] \to \mathbb{R}$ controls the state magnitude reduction per iteration, similar to residual connections in neural networks[26], where $T$ is the number of timesteps in the annealing schedule. The factor $\beta$ determines the matrix–vector product scale, where the matrix $W$ encodes neural network weights or an optimization problem, and $f: \mathbb{R}^N \to \mathbb{R}^N$ is an element-wise nonlinear function. The coefficient $\gamma \in (0, 1)$ introduces momentum which, in continuous time, corresponds to the second-order differential equation dynamics (Supplementary Information section G.10), generalizing the AOC abstraction beyond first-order models such as Hopfield networks. The bias vector $\mathbf{b} \in \mathbb{R}^N$ represents additional problem-specific information.

The introduced fixed-point abstraction is ideally suited for analog feedback devices, such as the AOC, as it is compute-bound, requires no intermediate memory and is noise-tolerant: the attracting fixed point pulls the trajectory closer at every iteration, counteracting analog noise.

## Hardware

The AOC hardware combines 3D optical and analog electronic technologies to accelerate all the compute operations in the fixed-point abstraction described by equation (1): matrix–vector multiplication, nonlinearity, annealing, addition and subtraction. In each fixed-point iteration, the analog signal alternates between optical and electrical domains, giving the system state $\mathbf{s}_t$ a dual opto-electronic nature.

Matrix–vector multiplication occurs in the optical domain, where the state vector $\mathbf{s}_t$ is encoded in the light intensity of arrays of micro light-emitting diodes (microLEDs), whereas the weight matrix $W$ is represented by spatial light modulator (SLM) pixels (Fig. 1a). Light from each microLED fans out across an SLM row for element-wise multiplication, and the resulting light signals are summed column-wise by a photodetector array[27]. In contrast to planar optical architectures, the AOC leverages 3D optics with its efficient fan-in and fan-out of light in the third dimension through the use of spherical and cylindrical optics, thus enabling inherently parallel and scalable multiplication operations of larger matrices[28].

The result of the optical matrix–vector multiplication is measured in the electrical domain using a photodetector array, where the state vector $\mathbf{s}_t$ is represented as a voltage per detector. The remaining operations in equation (1) are implemented via analog electronics: a hyperbolic tangent (tanh) function for the element-wise nonlinearity, summing and difference amplifiers for addition and subtraction of analog signals, and variable gain amplifiers for the annealing schedule $\alpha(t)$ and the $\beta$ factor. The circuit layout, with highlighted voltage readout position, is detailed in Extended Data Fig. 3.

The AOC hardware executes the iterative update rule from several to thousands of iterations until convergence to a fixed point, when the signal amplitudes are read out digitally. This all-analog operation minimizes the overhead of analog-to-digital conversions. The current hardware includes 16 microLEDs and 16 photodetectors, supporting a 16-variable state vector $\mathbf{s}_t$, along with two SLMs to handle positive and negative entries of the matrix $W$ (Fig. 1b,c). This configuration is sufficient to support inference for ML models and optimization tasks with up to 256 weights and can extend to 4,096 weights via problem decomposition, maintaining fully analog fixed-point iterations.

The AOC fixed-point abstraction targets a balance between generality and efficient hardware implementation. To illustrate its versatility, the remainder of the paper presents four case studies highlighting how equilibrium ML models can be applied to classification and regression tasks, and how the QUMO paradigm can represent real-world applications in finance and healthcare, while utilizing the same AOC hardware.

## AOC for machine learning
### Analog equilibrium model

The AOC supports neural equilibrium models, which have been widely applied across various domains from language[17] to vision[29]. Equilibrium models typically follow a fixed-point iterative update rule, $\mathbf{s}_{t+1} = \text{Network}(\mathbf{s}_t)$, with examples including classic Hopfield networks[30] and their modern variants[31], as well as deep-equilibrium models[17]. These models operate as self-recurrent neural networks with constant input, driving the hidden state to a fixed point that represents the network output (Fig. 1d). Their dynamic depth enables recursive reasoning, leads to improved scaling laws[32] and enhances out-of-distribution generalization compared with feedforward models[18,33]. Recent self-recurrent language models with billions of parameters show impressive reasoning capabilities, surpassing fixed-depth models in representation power[34,35]. We demonstrate that the AOC supports models with such recurrent nature and achieves greater out-of-distribution generalization (Extended Data Fig. 6).

As illustrated in Fig. 2a, the complete neural network architecture includes an input projection (IP) layer, the equilibrium model and an output projection (OP) layer. The network training is performed digitally using the AOC-DT, whereas the equilibrium model is deployed on the AOC hardware for inference.

For equilibrium models, the fixed-point iterative update follows from equation (1) by setting $\alpha(t) \to \alpha$, $\gamma(t) \to 0$, $\mathbf{b} \to \mathbf{b} + \mathbf{x}_{\text{proj}}$, and using the element-wise tanh nonlinearity:

$$\mathbf{s}_{t+1} = \alpha\mathbf{s}_t + \beta W \tanh(\mathbf{s}_t) + \mathbf{b} + \mathbf{x}_{\text{proj}}. \tag{2}$$

Here $\mathbf{b}$ is the trained bias and $\mathbf{x}_{\text{proj}}$ encodes the equilibrium model input. The trained weight matrix $W$ is quantized to 9-bit integers (Methods), separated into positive and negative components, and loaded into the corresponding SLMs. The initial state is set to $\mathbf{s}_0 = \mathbf{b} + \mathbf{x}_{\text{proj}}$.

During inference, the original data $\mathbf{x}$ go through the IP layer as $\mathbf{x}_{\text{proj}} = W_{\text{IP}}\mathbf{x} + \mathbf{b}_{\text{IP}}$, where $W_{\text{IP}}$ and $\mathbf{b}_{\text{IP}}$ are the trained IP weights and biases. For the given $\mathbf{x}_{\text{proj}}$, the equilibrium model iterates on the AOC hardware until convergence, with the fixed-point state $\mathbf{s}^*$ read out as voltages (Extended Data Fig. 5). Finally, the inference result is obtained by applying the OP layer to the AOC solution as $\mathbf{y} = W_{\text{OP}}\mathbf{s}^* + \mathbf{b}_{\text{OP}}$, where $W_{\text{OP}}$ and $\mathbf{b}_{\text{OP}}$ are the trained OP weights and bias.

In the current hardware, the equilibrium model is implemented for a single-layer network with a 256-weight matrix, without symmetry constraints. A recurrent multilayer neural network can be constructed using a lower subdiagonal block-wise matrix (Fig. 3), whereas a continuous-valued Hopfield network arises for a symmetric matrix. We note that neural networks conventionally apply the activation function and weight matrix in a different order than in equation (2). However, this leads to only a minor difference in practice owing to the self-recursive process[36].

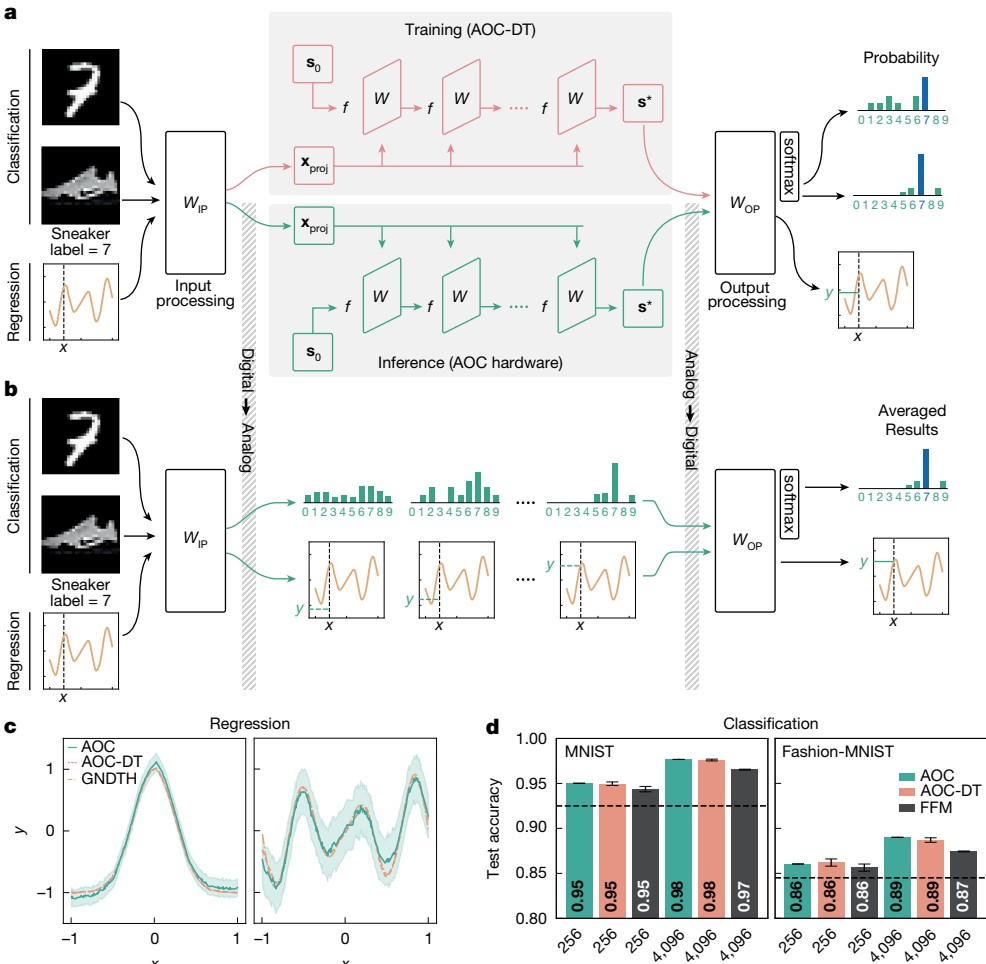

**Fig. 2 | AOC for ML inference. a**, Top: the neural network architecture used for training and inference. During training, input data are projected into the latent space via an IP layer, processed by the AOC hardware's digital twin (AOC-DT) and passed through the OP layer. Bottom: the inference process on the AOC hardware: the IP and OP layers interface with the hardware, which updates its initial state $\mathbf{s}_0$ until reaching the fixed-point state $\mathbf{s}^*$. **b**, During inference, MNIST and Fashion-MNIST images are passed through the IP layer and are fed into the AOC hardware. Intermediate states may be projected out to monitor progress, schematically showing evolution from a higher to a lower entropy distribution. Once converged, a softmax nonlinearity in the OP layer selects the highest-probability class, whereas in regression, the OP layer outputs a continuous value $y$, with the MSE evaluated against ground truth (GNDTH) to assess performance. GNDTH is the true curve we want to regress against. **c**, The nonlinear regression results are demonstrated for the AOC hardware over a Gaussian curve (left) and sinusoidal curve (right) with MSE losses of $3.75 \times 10^{-3}$ and $1.21 \times 10^{-2}$, respectively. The shaded area around the AOC predictions represents the observed variability, that is, the standard deviation across the sampling window and repeated AOC runs with the same input (Methods). **d**, For the full test datasets of MNIST (left) and Fashion-MNIST (right), the AOC classification accuracy is compared with the performance of the AOC-DT and the feedforward model (FFM). Higher accuracies are achieved for larger models (4,096 weights), which are realized with a time-multiplexing technique. Hardware results align with AOC-DT simulations; the 4,096-weight hardware result slightly exceeds the AOC-DT, as it reflects the best of 2 seeds, whereas the AOC-DT accuracy is averaged. The dashed lines show the linear classifier performance; the error bars reflect the random-seed variability for the AOC-DT and experimental repeats for hardware (Methods).

The hardware realizes the equilibrium model as in equation (2) for two ML inference tasks: image classification and nonlinear regression (Fig. 2b). In both cases, models are trained digitally through the AOC-DT and are deployed on the hardware without further calibration, which requires high hardware precision and high fidelity of the AOC-DT (Methods).

## Regression case study

We show that the AOC hardware can run nonlinear regression models. Regression tasks require continuous-valued outputs, which are challenging owing to the inherently noisy nature of analog computations. This is in contrast to classification tasks discussed below where the output labels are discrete and only the class of the largest probability is selected. From a model perspective, nonlinear regression tasks are well suited to the 256-weight AOC as they require small input ($W_{IP} \in \mathbb{R}^{16 \times 1}$) and output ($W_{OP} \in \mathbb{R}^{1 \times 16}$) projection matrices.

We select two nonlinear functions for regression: Gaussian and sinusoidal curves. In agreement with the AOC-DT results (see Supplementary Table 2), the hardware reproduces accurately both functions, as shown in Fig. 2c. The sinusoidal curve presents a greater challenge for accurate fitting than the Gaussian curve owing to its multiple minima and maxima, requiring higher AOC-DT fidelity. This may explain why the AOC hardware struggles to accurately fit the region near the right minimum of the sinusoidal curve. However, at no point does the AOC-DT curve fall outside the AOC standard deviation. We note the digital IP and OP layers alone would only be able to fit linear functions, highlighting the contribution of the equilibrium model running on the AOC hardware.

## Classification case study

For the Modified National Institute of Standards and Technology (MNIST) and Fashion-MNIST datasets, the inputs $\mathbf{x} \in \mathbb{R}^{28 \times 28}$ are rescaled to $[-1, 1]$ range and flattened into vectors. Hence, the IP and OP layers

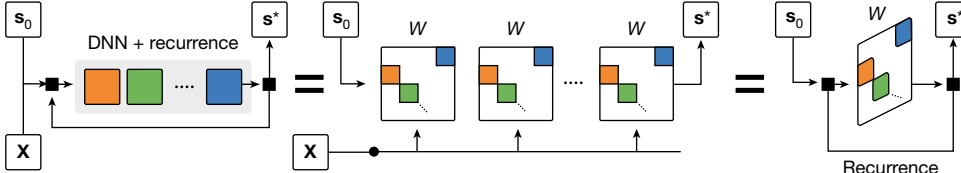

**Fig. 3 | Multilayer neural networks with recurrence on the AOC.** The left panel shows a multilayer deep neural network (DNN) with recurrence, where orange, green and blue denote different layers. Here, $\mathbf{s}_0$ is the initial network state, $\mathbf{x}$ is the network input and $\mathbf{s}^*$ represents the fixed point to which the network converges. This network structure can be realized on the AOC hardware by arranging the layers along the lower subdiagonal of a larger matrix $W$, placing the final layer as a recurrent block in the top-right corner, with its time evolution shown in the middle panel. The right panel further represents this architecture through a single block iterated over time, capturing the recurrent multilayered model structure. All three panels depict equivalent representations of the recurrent multilayer network.

have dimensions $16 \times 784$ and $10 \times 16$, respectively. The test dataset results on the 256-weight AOC hardware are shown in Fig. 2d (see also Supplementary Table 4). The AOC-predicted labels match the AOC-DT results for 99.8% of the inputs.

The AOC results demonstrate the viability of digital training with subsequent weight transfer for opto-electronic analog inference. The contribution of the equilibrium model running on the AOC further becomes apparent when comparing the AOC results with a linear classifier, which consists of digitally trained IP layer, a middle layer and an OP layer. We also train a simple feedforward model comprising an IP layer, a middle layer with tanh nonlinearity and an OP layer. Both linear classifier and feedforward models have the same number of parameters as the AOC hardware. Although the AOC achieves slightly higher accuracy (Fig. 2d), the simple nature of the MNIST and Fashion-MNIST datasets is unlikely to demonstrate the full potential of self-recurrent models. Looking ahead, this potential may materialize in a form of test-time compute in sequence modelling tasks[34,35] or inference of generative diffusion models.

In practice, model sizes tend to exceed what a given hardware can support, including traditional GPUs. To address this, we demonstrate time-multiplexing on the AOC by training a 4,096-weight ensemble equilibrium model, composed of 16 independent 256-weight equilibrium models. The overall architecture mirrors the previous 256-weight model, but the middle layer now consists of 16 independent equilibrium models, executed sequentially on the AOC for each slice of the input $\mathbf{x}_{\mathrm{proj}} \in \mathbb{R}^{4096}$. Classification accuracies for these time-multiplexed models are shown in Fig. 2d. The time-to-solution increases linearly with the number of independent blocks in the ensemble. Across all architectures considered for classification tasks, the IP layer accounts for the majority of parameters, whereas nonlinearities within the AOC primarily drive performance differences compared with a linear classifier. Additional classification results, including ablation studies of the optical and electronic contributions with untrained IP layers, are provided in Supplementary Information section C.1.

Across all classification and regression tasks, the AOC-DT requires around nine iterations per input to reach convergence. On the AOC hardware, these fixed points can be achieved in 180 ns and, ideally, their sampling would occur immediately afterwards. In practice, to ensure the state stability and mitigate noise, we sample over a fixed 6.4-µs window at 6.25-MHz frequency (Extended Data Fig. 5). As the sampling rate is about eight-times slower than the hardware round-trip time, individual iterations cannot be resolved on the AOC. Compared with classification tasks, the regression tasks show greater sensitivity to noise, requiring up to 11 repeated runs for averaging to obtain smooth curves (Supplementary Fig. 6). We note that owing to the iterative process, the training and inference times of the AOC-DT running in silico are approximately nine-times slower than an equivalent feedforward model.

## Noise robustness

Equilibrium models, beyond being compute heavy, are also suitable for analog acceleration owing to the attractor nature of their iterative inference process. This provides enhanced robustness to analog noise compared with deep feedforward networks, which is a critical property for the AOC performance at scale (Extended Data Fig. 6b).

## AOC for optimization
### Quadratic unconstrained mixed optimization

The QUMO formulation represents a wide class of combinatorial optimization problems aimed at minimizing the objective function $F(\mathbf{x}) = -\frac{1}{2}\mathbf{x}^{\mathrm{T}}W\mathbf{x} - \mathbf{b}^{\mathrm{T}}\mathbf{x}$, where the vector $\mathbf{x}$ includes binary and continuous variables, and the information about the optimization problem is encoded in the weight matrix $W$ and the constant vector $\mathbf{b}$. Without loss of generality, one may consider the values $\{0, 1\}$ for binary and the interval $[0, 1]$ for continuous variables. These variables are represented through the element-wise nonlinearity over the system state vector $\mathbf{s}_t$ in equation (1). The solution to the QUMO problem is the assignment of the variables $\mathbf{x}$ that minimizes the objective $F(\mathbf{x})$. If the components of $\mathbf{x}$ are all binary variables, the problem reduces to the quadratic unconstrained binary optimization (QUBO) formulation. We note that the QUBO problem is equivalent to the well-known problems of minimizing the Hamiltonian of the Ising model[37] and finding the maximum cut of a weighted graph[38].

Besides being nonlinear, most optimization problems are constrained. A problem with linear inequality constraints highlights the greater expressiveness of the QUMO over the standard QUBO formulation, commonly used across many non-traditional platforms. For example, only one additional continuous variable, typically referred to as slack variable, is required for mapping one inequality constraint to the QUMO problem with a penalty method. In contrast, the QUBO formulation suffers from a large mapping overhead: 10 to 100 binary variables are needed to represent a single constraint with either binary or unary encoding (Supplementary Information section G.8). We next demonstrate solving QUMO problems on the AOC hardware for two applications: medical image reconstruction and transaction settlement.

### Medical image reconstruction case study

We implement compressed sensing on the AOC hardware, a technique enabling accurate signal reconstruction from fewer measurements than traditionally required[39,40]. Compressed sensing accelerates image acquisition, reducing scan times and enhancing patient comfort. For magnetic resonance imaging (MRI), a sparse image representation is typically achieved using techniques such as wavelet regularization that penalize 'unnatural' reconstructions. The standard regularization choice is the $\ell_1$-norm, which promotes sparsity and enables optimization via convex solvers. However, the original compressed-sensing method employs the '$\ell_0$-norm', which counts the number of non-zero elements in a vector. Minimizing the $\ell_0$-norm may yield better reconstruction in theory[41,42], although the optimization problem is deemed impractical in this case and, hence, remains largely unexplored in applied image reconstruction tasks. With the AOC hardware, we can address

this original hard problem by formulating the compressed-sensing approach as the QUMO optimization problem:

$$\min_{\mathbf{x}} \frac{1}{2} \|\mathbf{y} - \mathbf{A}\mathbf{x}\|_2^2 + \lambda_1 \mathbf{1}^{\mathsf{T}} \boldsymbol{\sigma} + \lambda_2 (\mathbf{1} - \boldsymbol{\sigma})^{\mathsf{T}} \mathbf{x}. \qquad (3)$$

Here the first term ensures data fidelity between measurements $\mathbf{y} \in \mathbb{R}^M$ and the image $\mathbf{x} \in \mathbb{R}^N$ in the wavelet domain, and the matrix $\mathbf{A} \in \mathbb{R}^{M \times N}$ represents the MRI acquisition process consisting of Fourier and inverse wavelet transforms with an undersampling mask (Methods). To reduce the MRI scan time, the number of measurements $M$ needs to be smaller than the number of pixels $N$; hence this data-fidelity term has infinitely many solutions on its own. The image pixels are normalized to $\mathbf{x} \in [0, 1]^N$ in the wavelet domain and $\boldsymbol{\sigma} \in \{0, 1\}^N$ is a binary vector that controls the sparsity of $\mathbf{x}$. When $\sigma_i = 0$, the $\lambda_1$ penalty disappears and the corresponding non-zero pixel value $\mathbf{x}_i$ penalizes the objective owing to the $\lambda_2$ penalty. For $\sigma_i = 1$, the $\lambda_2$ penalty disappears and the pixel $\mathbf{x}_i$ can take any value to match the measurements, albeit at $\lambda_1$ penalty cost to the objective. Lastly, $\mathbf{1}$ and $(\cdot)^{\mathsf{T}}$ denote the vector of ones and the transpose operation in equation (3), respectively. The generalization of this reconstruction problem to the complex-valued variables is presented in Supplementary Information section G.3.

We realize the compressed-sensing approach on the AOC hardware for a line of the Shepp–Logan phantom image of 32 × 32 pixels (Fig. 4a and Supplementary Fig. 10), formulated as a 64-variable 9-bit QUMO problem with equal split between binary and continuous variables. As an example of a realistic MRI process, we omit 37.5% of the measurements. By minimizing the data-fidelity term only, we obtain a poor reconstruction (Fig. 4a), highlighting the importance of the interactions between continuous and binary variables in the QUMO formulation.

For the non-zero $\lambda_1$ and $\lambda_2$ penalties, we split the 64-variable QUMO problem into smaller subproblems and solve them all in the AOC hardware using the block coordinate descent (BCD) method[43]. Solutions to each subproblem, corresponding to one step of BCD, are used to create the subsequent ones, all of which are solved using the fixed-point abstraction realized on the AOC hardware. The convergence to the optimal solution takes around 30 –40 BCD steps with 1,000–1,500 AOC iterations per BCD step. The final image reconstruction closely matches the original line, as shown in Fig. 4a. We note that all QUMO instances are solved in an entirely analog manner without any digital post-processing.

To validate the QUMO formulation for compressed sensing at scale, we use the AOC-DT to reconstruct a brain scan image of 320 × 320 pixels from the FastMRI dataset[44], which results in the QUMO problem with more than 200,000 variables. For the typical undersampling rates of 4 and 8, we achieve reconstructions with mean squared error (MSE) below 0.07 (Fig. 4a).

### Transaction settlement problem case study

For optimization in the financial domain, we use the AOC hardware to solve a transaction-settlement problem. Each securities transaction is an exchange of securities for a payment, known as a delivery-versus-payment transaction. Clearing houses process batches of such transactions; for example, the subsidiaries of the Depository Trust and Clearing Corporation (DTCC) processed securities transactions valued at US$3 quadrillion in 2023[45]. Within each batch, the transaction-settlement objective is to maximize the total number or total value of settled transactions, which is NP-hard[46]. This is a difficult optimization problem given the volume of transactions, legal constraints and additional requirements (for example, collateral and credit facilities).

A prevalent approach for solving the transaction-settlement problem is to formulate it as a linear optimization problem with binary variables and linear inequality constraints[46]. This formulation can be mapped to a QUMO problem, where the inequality constraints are efficiently incorporated into the objective function by introducing continuous slack variables (Supplementary Information section G.8).

We design a transaction-settlement instance generator that produces industrially relevant transaction-settlement scenarios for the given numbers of transactions, financial parties and assets. We also implement a pre-processing technique to eliminate trivial constraints. As an example, we generate a scenario with 46 transactions between 37 parties (Fig. 4b), resulting in 30 constraints, which is reduced to an effective 41-variable QUMO instance. As shown in Fig. 4b, the AOC hardware finds the globally optimal solution in seven BCD steps for this transaction-settlement scenario. Similar to reconstruction of the Shepp–Logan image, the QUMO instances across all transaction-settlement scenarios are solved in an entirely analog manner.

In addition, we evaluate several smaller-size scenarios derived from real settlement data[47]. After pre-processing, these reduce to QUMO instances of 8 variables, on which the AOC hardware achieves a 100% success rate (see Supplementary Table 8). In contrast, quantum hardware performance on the same problems yields success rates of 40–60% (ref. 47).

### Comprehensive benchmarking

For the AOC hardware, a comprehensive evaluation is conducted on a diverse set of challenging synthetic QUMO and QUBO problems. The optimization problems include instances with 16 binary and continuous variables, with dense and sparse weight matrices up to 8-bit precision. For 100 instances, the AOC hardware achieves over 95% and 100% proximity to the optimal objectives of QUMO and QUBO instances, respectively, under 1,000 samples (Fig. 4c). The typical time trace of 16 variables during the optimization process on the AOC hardware is shown in Fig. 4d.

Using the AOC-DT, algorithmic performance is validated on the hardest quadratic binary problems with linear inequality constraints from the quadratic programming library (QPLIB) benchmark[23], formulated as QUMO instances. The AOC approach is compared with the commercial Gurobi solver[22], which requires over a minute to reach the best-known solutions for these problems. Figure 4e,f shows that the AOC-DT is up to three orders of magnitude faster in all instances, except for two, one of which it is unable to solve. Moreover, the AOC solver discovers the new best solutions for two heavily constrained instances (3,584 and 3,860), each with over 500 binary and 10,000 continuous variables in the QUMO formulation, in about 40 s. For instance, for 3,584, Gurobi matches the AOC solution in about 54,000 s, whereas proving its global optimality takes 4.5 days. The details of the AOC-DT parameters are provided in Supplementary Information section G.4 with additional benchmarks in Supplementary Information section G.5.

### Discussion

Addressing practical applications with the AOC necessitates hardware scalability from hundreds of millions to billions of weights. For example, typical MRI scans with resolutions around 100,000 pixels require systems capable of processing around 20,000 variables when using decomposition techniques such as BCD, which is equivalent to handling around 400 million weights. Similarly, deep learning models with a few billion weights are standard for real-world applications[48], which, with mixture of experts models or techniques such as ensembling, could be reduced to many parallel models that are an order of magnitude smaller. The AOC hardware has the potential to scale to these requirements through a modular architecture that decomposes the core optical matrix–vector multiplication operation into multiplication of smaller subvectors and submatrices.

At scale, the AOC hardware will consist of multiple modules, each performing a part of the full-weight matrix multiplication. Each module will include a microLED array, a photodetector array and an SLM.

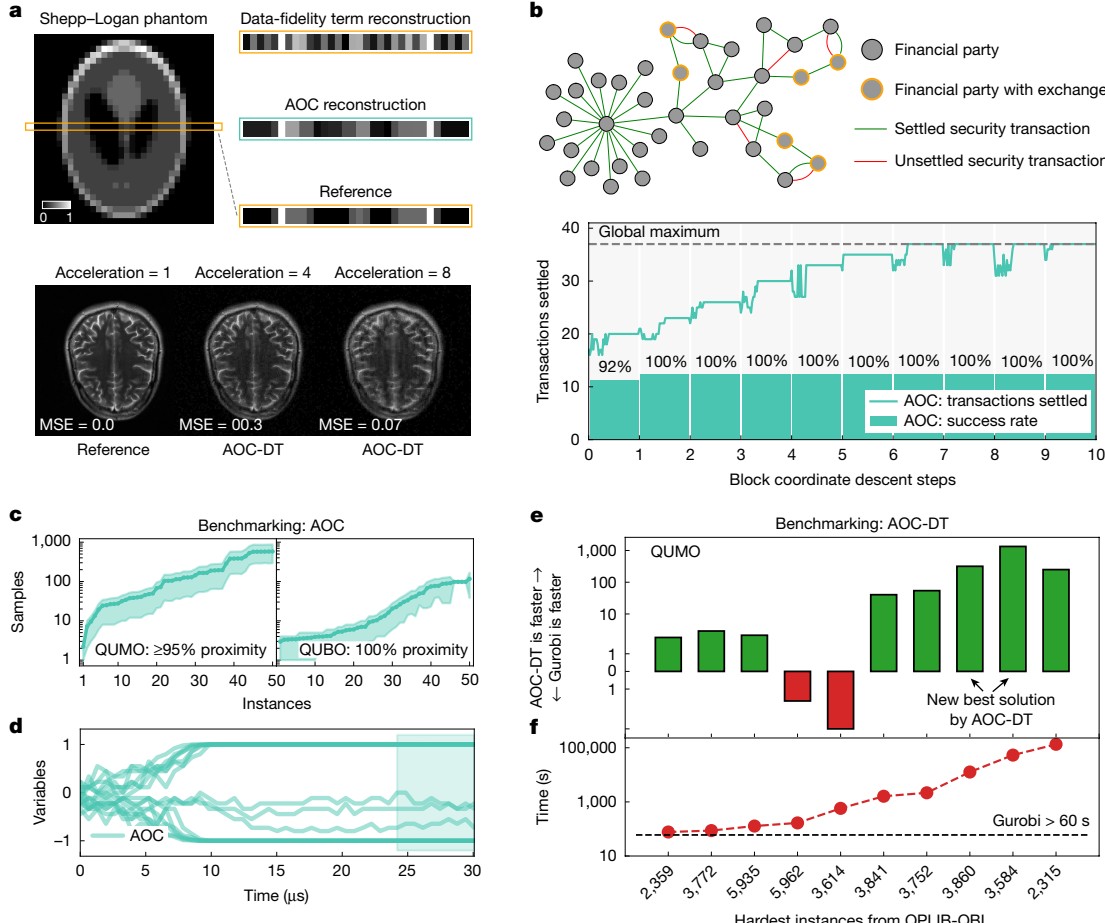

**Fig. 4 | The AOC for optimization. a**, Medical image reconstruction. Top: the AOC hardware realizes a compressed-sensing algorithm with the '$\ell_0$-norm' to reconstruct a line of the Shepp–Logan phantom image. The reconstruction process is formulated as optimization of the 64-variable QUMO instance and compared with the minimization of the data-fidelity term only in equation (3), achieving an MSE of 0.008 in the former and 0.079 in the latter cases. The greyscale bar denotes brightness values in the image, with 0 corresponding to black and 1 to white. Bottom: reconstructions by the AOC-DT for a real brain image from the FastMRI dataset at typical acceleration rates. **b**, Transaction settlement. Top: a schematic of the transaction-settlement process among multiple financial parties, with settled (green) and unsettled (red) transactions. Bottom: the number of settled transactions achieved by the AOC hardware for a 41-variable QUMO instance as a function of block coordinate descent steps.

**c**, The AOC hardware solves synthetic 3-bit to 8-bit precision QUMO and QUBO instances with 16 variables, requiring fewer than 1,000 samples to reach at least ≥95% and 100% objective proximities for the QUMO and QUBO instances, respectively. The shaded regions indicate the 50% confidence interval. **d**, The time trace of 16 variables during the optimization process on the AOC hardware for a QUMO instance. The system converges to the fixed point within 30 µs, with the sampling window occurring over the last 6 µs. **e**, The relative speed-up of the AOC-DT compared with the Gurobi solver is shown for a QUMO-reformulated subset of QPLIB benchmark instances (QBL). The state-of-the-art solutions are found for instances 3,860 and 3,584. **f**, The hardest QBL instances require more than 60 s for the Gurobi solver to find the best-known solutions of 10 QBL instances in their original formulation.

With current SLMs supporting 4 million pixels, matrices with up to 4 million weights could be realized. When combined with integrated driving electronics, this results in a miniaturized module with dimensions of around 4 cm. Utilizing the third dimension for matrix–vector multiplication thus enables scalable in-memory computing. In contrast, emerging planar optical computers[1,2] are constrained by the reticle-size limit of the chip area that is used for both routing and computing, which limits the matrix size[28]. Furthermore, as microLEDs are incoherent light sources, optical paths need to be matched only within the system bandwidth (gigahertz) rather than the source wavelength (hundreds of terahertz), which is a fundamental manufacturability advantage over coherent systems. Achieving the required miniaturization, however, is both a challenge and an opportunity to drive advancements in 3D optical technologies with broader applications. The miniaturized optical modules are coupled with integrated analog electronic models in a 3D mesh (Extended Data Fig. 4). These modules further aggregate the output vectors from the optical modules and perform the remaining compute primitives.

We envision that the AOC can support models with 0.1 billion to 2 billion weights, requiring 50 to 1,000 optical modules. The module count can be halved if a single optical module supports both positive and negative weights[13]. All AOC components, including microLEDs, photodetectors, SLMs and analog electronics, have an existing and growing manufacturing ecosystem with wafer-scale production. At the same time, complementing optics with analog electronics offers numerous opportunities to expand the compute primitives, including nonlinearities, the hardware can support, thereby enhancing its expressiveness.

The operational speed and power consumption of the AOC dictate its energy efficiency. The speed is limited by the bandwidth of the opto-electronic components, typically 2 GHz or higher[49]. For a 100-million-weight matrix with 25 AOC modules, the power consumption is estimated to be 800 W, resulting in a computing speed of 400 peta-OPS and an efficiency of 500 TOPS W$^{-1}$ (2 fJ per operation) at 8-bit weight precision (Supplementary Information section A.1). In contrast, the latest GPUs achieve a system efficiency of up to 4.5 TOPS W$^{-1}$ at the same precision for dense matrices[24].

In conclusion, the AOC architecture shows promise for scaling to practical ML and optimization tasks, offering a potential 100-fold improvement in power efficiency. The current AOC hardware uses a rapid fixed-point search to power inference tasks, such as regression and classification, using equilibrium models with promising reasoning capabilities, and to successfully solve QUMO problems including medical image reconstruction and transaction settlement. Cross-validation with the digital twin, coupled with evaluation on large problems, offers confidence in the hardware's performance as it scales. Looking ahead, the AOC's co-design approach—aligning the hardware with the ML and optimization algorithms—could spur a flywheel of future innovations in hardware and algorithms, pivotal for a sustainable future of computing.

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

## Methods

### Experimental set-up

The key components of our experimental set-up are shown in Fig. 1a and Extended Data Fig. 1.

**Optical subsystem.** The optical subsystem performs matrix–vector multiplication. The basic components are the optical sources (input vector), a system of fan-out optics to project the light onto the modulator matrix and a system of fan-in optics to project the light onto a photodetector array (output vector). The corresponding schematic is shown in Extended Data Fig. 2.

The incoherent light sources are an array of 16 independently addressable microLEDs. Each microLED is driven with a bias current and an offset voltage. The variable value is encoded by the light intensity, with a value of zero corresponding to the microLED bias point. Mathematical positive values are represented by microLED drive currents greater than the bias value. Negative values are represented by drive currents less than the bias value. The diameter of each emitter is 50 µm and the pitch is 75 µm. The sources are fabricated in gallium nitride wafers on a sapphire substrate and the die is wire-bonded onto a printed circuit board (Fig. 1c). The emission spectrum is centred at 520 nm with a full-width of half-maximum of 35 nm and the operational −3-dB bandwidth is 200 MHz at 20 mA, see Supplementary Fig. 1.

After the sources, there is a polarizing beamsplitter (PBS). From this point, there are two equivalent optical paths in this set-up. Each path performs two functions: first, they allow us to use both polarizations of the unpolarized light output; second, they allow us to perform non-negative and non-positive multiplications with only intensity modulation. Each path contains one amplitude modulator matrix and one photodetector array. The modulator matrix is a reflective parallel-aligned nematic liquid-crystal SLM. We refer to the first part of the optical system as the fan-out system. The task of this fan-out system is to image the microLEDs onto the SLM, where the weights are displayed, and to spread the light horizontally into lines. The microLEDs are arranged in a one-dimensional line (let this be the $y$ axis) and are imaged onto the SLM using a 4F system composed of a high-numerical-aperture (Thorlabs TL10X-2P, numerical aperture 0.5, ×10 magnification, 22-mm field number) collection objective and a lower-numerical-aperture lens group composed of 2 achromatic doublets with combined focal length 77 mm. There is a cylindrical lens, Thorlabs LJ1558L1, in infinity space of this 4F system. This lens adds defocus to the image of the source array on the SLM but only in the $x$ direction, so that the projected light pattern is a set of long horizontal lines, one per microLED. Each matrix element occupies a patch of 12 (height) × 10 (width) pixels of the modulator array. An 8-bit look-up table is used to linearize the SLM response as a function of grey level.

The SLM is imaged onto the photodetector array using a 4F system (the fan-in system). The first lens group of the fan-in is the same as the second lens group of the fan-out system as this is in double pass. From here, the light is directed towards the intended photodetector array through a second PBS. The light from each column of the SLM is collected by an array of 16 silicon photodetectors to perform the required summation operation. The active area of each element is $3.6 × 0.075$ mm$^2$. The photodetectors are on a pitch of 0.125 mm. The operation bandwidth is 490 MHz at −10 V measured at 600 nm.

**Analog electronic subsystem.** After the photodetector array, the signals are in the analog electronic domain. The photocurrents from each photodetector element are amplified by a linear trans-impedance amplifier (Analog Devices MAX4066). Each trans-impedance amplifier provides 25-kΩ gain and is characterized by an input referred noise of 3 pA $\sqrt{Hz}$ and has differential outputs. The corresponding 2 sets (1 per photodetector board) of 16 differential pairs of signals are fed to the main boards where the per-channel nonlinear operation and other

analog electronic processing is carried out. Each of the 16 signals sees the following circuitry: (1) a variable gain amplifier (VGA; Texas Instruments VCA824) to allow the input signal range to be set and equalized across channels; (2) a difference amplifier to perform the operation of subtracting the negative input signal from the positive one and achieve signed voltages (signed multiplications); (3) a VGA that adds and subtracts signals from the described path, referred to as gradient term, to the annealing and momentum terms, as per equation (1), while providing a common gain control to all these paths; (4) an electronic switch (ADG659) to open and close the loop to set and reset the solving state; (5) a buffer amplifier to distribute the signal to the gradient, annealing and momentum paths; (6) a bipolar differential pair to implement the tanh nonlinearity; (7) a VGA to adjust the signal level between the nonlinearity and the required voltage and current onto the microLED alternating-current input circuit. Both the annealing and momentum paths have VGAs with a common external control so that we can implement time-varying annealing and momentum schedules.

Each channel also has an offset to the common control signal added to allow minor adjustment or correction of channel-to-channel variations. The other VGAs are set with digital-to-analog converters controlled over an inter-integrated circuit (I2C) bus. This allows slower control at per-experiment timescales.

**Nonlinearity.** The per-channel nonlinear function is an approximation to a tanh. This is shown in Supplementary Fig. 5d. The system is designed so that all signals follow the same path through the solver. For ML workloads, the input domain of the tanh function is unrestricted by hardware; there are no gain variations across channels. The trained weights and equilibrium model input ensure that signals evolve accurately. For optimization workloads, binary and continuous variables require different handling in hardware. Here we set the gain after the trans-impedance amplifier and before the tanh nonlinearity to be lower for continuous variables than for binary variables. This adjustment ensures that the input domain of the nonlinearity results in a more linear output for continuous variables than for binary variables.

**Evaluation of matrix–vector multiplication accuracy.** We characterized and calibrated the key opto-electronic and electronic components to equalize the gain of each AOC path. For example, we calibrate the optical paths by applying a set of 93 reference matrices and for each we digitally compute the result of the vector–matrix product. We then adjust the gain per channel slightly so that, averaged over the set of 93 computed vectors, the AOC result is as close as possible to the digital result.

Following this, the accuracy of the matrix–vector multiplication is characterized using the same 93 reference matrices on each SLM and measuring the output of the system, shown in Supplementary Fig. 3a. For each reference matrix in the set, we calculate the MSE between the known and the measured output. The mean MSE across all dot products is $5.5 × 10^{-3}$, and the matrix–vector multiplication MSE as a function of matrix (instance) is shown in Supplementary Fig. 3b. For these experiments, we configure the system in open-loop mode without feedback and turn off the annealing and momentum paths.

### ML methods

**Training and digital twin.** In commercial deployments, training consumes less than 10% of the energy and, hence, is not targeted by the AOC. The equilibrium models are trained through our digital twin, which is based on equation (2). In the digital domain during training, the convergence criterion is set to $||\mathbf{s}_{t+1} - \mathbf{s}_t|| < \varepsilon$, with $\varepsilon = 10^{-3}$. The AOC-DT models up to seven non-idealities measured on the AOC device; each non-ideality can be switched on and off (Supplementary Fig. 5). The AOC-DT is implemented as a Pytorch module with the weight matrix $W$ and bias terms $b$, as well as the gain $\beta$ as trainable parameters. The weight matrix is normalized to fulfil $||W||_\infty = 1$ throughout training to

simulate the passive SLM. The numeric scale of the matrix is instead modelled by the gain $\beta$. This separation of scale is necessary as several nonlinear non-idealities occur between the matrix multiplication and the gain in equation (2), as discussed in Supplementary Information section D.

The weight matrix is initialized with the default Pytorch initialization for a $16 \times 16$ matrix, the bias term is initialized to 0 and $\beta$ is initialized at 1. We trained all models with a batch size of $B = 8$, at a learning rate of $\eta = 3 \times 10^{-4}$ for MNIST and Fashion-MNIST and $\eta = 7 \times 10^{-4}$ for regression tasks. We used the Adam optimizer[50]. In all cases, models are trained end-to-end, with the equilibrium-section trained through our AOC-DT using the implicit gradient method[17], which avoids storing activations for the fixed-point iterations. This decouples memory cost from iteration depth as intermediate activations do not need to be stored. In all experiments, the $\alpha$ gain in equation (2) is set to 0.5 to strike a balance between overall signal amplitude and speed of convergence. Low $\alpha$ values cause the signal to be too weak, resulting in a low signal-to-noise ratio (Supplementary Information section D).

**Inference and export to the AOC.** Once training has completed, the weight matrix $W$ is quantized to signed 9-bit integers using

$$W \approx \frac{\max(W)}{255} \text{ clamp} \left[ \text{round} \left( \frac{W}{\max(W)} \times 255 \right) \right]_{\min=-256}^{\max=255} \quad (4)$$
$$= \frac{\max(W)}{255} W_Q,$$

with the rounded and clamped matrix on the right-hand side being the quantized weight matrix $W_Q$. Whenever we report AOC-DT results, we report results obtained with the quantized matrix.

Exporting trained models to the AOC requires several further steps. First, the model inputs **x** and the bias term **b** need to be condensed into a single vector $\mathbf{b}_{AOC} = \mathbf{b} + \mathbf{x}$ followed by clamp to ensure the values fit into the dynamic range of the AOC device (Supplementary Information section D). Second, as the optical matrix multiplication is implemented using SLMs, elements of the weight matrix are bounded by one such that all quantization-related factors disappear. However, the original maximum element of the matrix $\max(W)$ needs to be re-injected, which we achieve via the $\beta$ gain in equation (2), approximately restoring the original matrix $W$.

The quantized matrix is split into positive and negative parts, $W_Q = W_Q^+ - W_Q^-$, and each part is displayed on its respective SLM.

**AOC sampling and workflow.** Each classification instance (that is, MNIST or Fashion-MNIST test image) is run once on the AOC, and the fixed point is sampled at the point marked in Extended Data Fig. 3 after a short 2.5-μs cooldown window after the switch is closed, as shown in Extended Data Fig. 5a,b. The sampling window extends over 40 samples at 6.25 MHz, corresponding to 6.4 μs. This ensures that the search of fixed points for the equilibrium models happens entirely in the analog domain. Once sampled, we digitally project the vector into the output space. For classification, the input is projected from 784 to 16 dimensions, the output is projected from 16 to 10 classes. The label is then determined by the index of the largest element in the output vector (argument-max). For regression tasks, the IP and OP layers transform a scalar to 16 dimensions and back, respectively. The MSE results in Fig. 2c were obtained by averaging over 11 repeats for each input. This means that we restart the solution process 11 times, including the sampling window, and average the resulting latent fixed-point vectors. Importantly, the solve-to-solve variability appears to be centred close to the curve produced by the AOC-DT, enabling us to average this variability out (Supplementary Fig. 6).

**The 4,096-weight ensemble model.** We can expand the model sizes supported by the hardware by using an ensemble of small models that

fit on it. These smaller 256-weight models are independent at inference time but are trained jointly by receiving slices 16-sized slices of a larger input vector and stacking their outputs before the OP. To scale to a 4,096-weight equilibrium model, we expand the input space from 16 to $16 \times 16 = 4{,}096$ dimensions and the output space from 10 to $10 \times 16 = 160$ dimensions. The IP matrix is consequently a $784 \times 4{,}096$-shaped matrix and the OP matrix is shaped $160 \times 10$. MNIST or Fashion-MNIST images are scaled to the range $[-1, 1]$ and, projected to 4,096 dimensions and split into 16 slices of 16 dimensions. Each of the 16 equilibrium models then runs its respective slice of input vectors to a fixed-point. Once all 16 models are run on the AOC, we concatenate outputs and project them into the 10-dimensional output space where the largest dimension determines the predicted cipher.

**Nonlinear regression.** The first curve (I) is a Gaussian rescaled such that the Gaussian curve approximately stretches from −1 to 1, $f_I(x) = 2e^{-x^2/2\sigma^2} - 1$ for $\sigma = 0.25$ and $x \in [-1, 1]$. The second curve (II) is given by $f_{II}(x) = \sqrt{|x|} \, \sin(3\pi x)$. For training sets, we choose 10,000 equidistant points $x_i$ in the range $[-1, 1]$ whereas for test regression datasets, we choose 200 points randomly $x_i \approx U([-1, 1])$.

**Error estimation.** For regression tasks, we concatenate the 40 samples from all 11 repeats and calculate the standard deviation per point on the curve.

**Classification datasets.** We trained the MNIST and Fashion-MNIST models on 48,000 images from their respective training set, validated on a set of 12,000 images and tested them on the full test set comprising 10,000 images.

**Error estimation.** For experimental results, the error bars in Fig. 2d were estimated using a Bayesian approach for the decision variable $c_t \in \{0, 1, \ldots, 9\}$ for each sample $t$ along the sampling window per image. We assume an uninformative prior $p(c_t) = \text{beta}(1, 1)$, which we then update with the observed number of correct decisions $n_{success}$ and failures $n_{failure}$ over the sampling window. The variance of the conjugate posterior of a beta distribution is given by $\text{Var}(c_t|n_{success}, n_{failure}) = \frac{(1 + n_{success})(1 + n_{failure})}{(2 + n_{success} + n_{failure})^2(3 + n_{success} + n_{failure})}$. We use this to estimate the variance and, by taking the square root, the standard deviation per input image. The dataset error bars are then estimated as the mean of the standard deviations over all members of the dataset.

## Optimization methods

**Positive and negative problem weights.** To address optimization problems involving positive and negative weights on the AOC hardware, QUMO instances without linear terms can have up to eight variables, which applies to both transaction-settlement scenarios and reconstruction of one-dimensional line of the Shepp–Logan phantom image. The weight matrices are unsigned in synthetic QUMO and QUBO hardware benchmarks; hence the AOC hardware can accommodate up to 16-variable instances in the absence of linear terms. Such instance size difference arises because, when both positive and negative weights are present, non-idealities in the dual-SLM configuration reduce the accuracy of matrix–vector multiplication. To mitigate this, a single SLM is used to process both positive and negative weights, effectively halving the number of variables per instance.

**Industrial optimization problems.** For the transaction-settlement scenario and the Shepp–Logan phantom image slice, their 41-variable and 64-variable QUMO instances are decomposed into smaller 7-variable QUMO instances. For each of these subinstances, the 7 variables are connected with the rest of the variables via a linear vector **b**, which is incorporated into the quadratic matrix $W$ via an additional binary variable. This decomposition is repeated for each subinstance and the linear

vector **b** is updated at the end of each BCD iteration to create the next QUMO instance. Such an approach yields 8-variable QUMO instances and a single SLM is used to represent their positive and negative matrix elements, with analog electronics handling their subtraction, which effectively utilizes the full 16-variable capacity available in hardware. The required number of BCD iterations varies depending on factors such as the initial random state of the optimization instance variables, the selection of variable blocks among subinstances, and the order in which they are optimized.

For the one-dimensional Shepp–Logan phantom image, 12 out of 32 measurements are omitted, corresponding to a 37.5% data loss or a 1.6 undersampling (acceleration) rate. Although typical MRI acceleration ranges from 2 to 8, this rate is used here owing to the image's non-smoothness at a 32-pixel resolution.

**Binary and continuous variables.** In the AOC, binary variables are encoded using a hyperbolic tangent function, whereas continuous variables utilize the near-linear region of the function, connecting optimization variables to state variables via $\mathbf{x} = f(\mathbf{s})$. In simulations at scale with the AOC-DT, linear and sign functions are used for continuous and binary variables, respectively.

**Hardware QUMO instances.** To ensure that some variables take indeed continuous values in the global optimal solution, we plant random continuous values and generate synthetic 16-variable QUMO instances. As the number of continuous variables increases for a given problem size, the problem instances become computationally easier to solve. Consequently, we consider instances with up to eight continuous variables.

**Hardware QUBO instances.** We generate up to 8-bit dense and sparse instances. The sparse instances belong to the QUBO model on three-regular graphs that are NP-hard[51], although NP-hardness does not imply that every random instance is difficult to solve. To make these instances more challenging to solve, we verify that their global objective minimizer is distinct from the signs of the principal eigenvector of the weight matrix[52].

**QPLIB benchmark.** The QPLIB is a library of quadratic programming instances[23] collected over almost a year-long open call from various communities, with the selected instances being challenging for state-of-the-art solvers. As described in the main part of the paper, we consider only the hardest instances within the QPLIB:QBL class of problems, which contains instances with quadratic objective and linear inequality constraints. The QPLIB:QCBO class of problems, which contains instances with quadratic objective and linear equality constraints, and the QPLIB:QBN class of problems, which contains QUBO instances, are considered in Supplementary Information section G.5.

**AOC-DT operation and parameters.** The distinction of the AOC-DT algorithm is the simultaneous inclusion of both momentum and annealing terms, which markedly improves the performance of the standard steepest gradient-descent method on non-convex optimization problems. Typically, multiple hyperparameters need to be calibrated for heuristic methods to achieve their best performance in solving optimization problems. We consider $\alpha(t) = 1 - \hat{\alpha}(t)$, where $\hat{\alpha}(t)$ is a linearly decreasing function from some initial value $\alpha_0$ to 0 over time. From the hardware perspective, such an annealing schedule provides an explicit stopping criteria, which is an advantage for an all-analog hardware implementation as it avoids the complexity of multiple intermediate readouts that stochastic heuristic approaches suffer from[53]. In principle, the three main parameters $\{\alpha_0, \beta, \gamma\}$ of the AOC fixed-point update rule need to be adjusted for each optimization instance. In our simulations, we notice that the algorithm is less sensitive to the momentum parameter value, whereas the $\alpha_0$ and $\beta$ values substantially affect the solution quality. We further perform a linear stability analysis of the algorithm to evaluate reasonable exploration regions for these two parameters and find that by scaling the $\beta$ parameter as $\beta = \beta_0 / \lambda_{largest}$, where $\lambda_{largest}$ is the largest eigenvalue of the weight matrix $W$, we get scaled parameters $\beta_0$ and $\alpha_0$ being in a similar optimal unit range across a wide range of problems.

We design a two-phase approach for the AOC-DT to operate similar to a black-box solver that can quickly adjust the critical parameters within the given time limit. During the 'exploration' phase, we evaluate the relative algorithm performance across a vast range of parameters $(\alpha_0, \beta_0)$. A subset of 'good' parameters is then passed for more extensive investigation in the 'deep search' phase (Supplementary Information section G.1).

We note that for two QPLIB:QUMO instances, namely, 5,935 and 5,962, we developed a pre-processing technique that greedily picks variables with the highest impact on the objective functions and considers their possible values, which is accounted in the reported time speed-up of the AOC-DT.

**Competing solvers.** For a fair comparison, we ensure that all methods use similar computing resources. Although the implementation of GPU- or central-processing-unit-based solvers can require highly varying engineering efforts, we try to estimate the cost of running solvers on the hardware, on which they are designed to run, and vary the time limit across solvers accordingly to ensure similar cost per solver run. In what follows, the Julia-based AOC-DT runs on a GV100 GPU for 5–300 s per instance across all benchmarks. In the case of Gurobi, our licence allows us to use only up to eight cores, and its time to achieve the best solution for the first time is used (not the time to prove its optimality).

More details about the AOC hardware and the AOC-DT performance on different optimization instances are provided in Supplementary Information section G.5.

## Data availability
All data supporting the findings of this study are available in source data provided with the paper and via Zenodo at https://doi.org/10.5281/zenodo.15088326 (ref. 54). Source data are provided with this paper.

## Code availability
Code for the AOC digital twin supporting optimization and machine learning models will be released upon publication at https://micro-soft.github.io/AOCoptimizer.jl and https://github.com/microsoft/aoc (under MIT license).

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

**Acknowledgements** We acknowledge G. Mourgias-Alexandris and I. Haller for contributions to the first-generation AOC hardware; S. Jordan, B. Lackey, A. Barzegar, F. Hamze and M. Troyer for discussions about optimization problems and for providing us with the QUBO benchmarks (Wishart, Tile3D, RCDP); J. Cummins for contributions to medical image reconstruction; A. Grace for contributions to the AOC hardware; J. Westcott, N. Farrell and T. Burridge for the AOC mechanical parts; S. Y. Siew for help with microLEDs; M. Schapira and colleagues at Microsoft Research, Cambridge, UK, and S. Bramhavar from ARIA for discussions. N.G.B. acknowledges the support from HORIZON EIC-2022-PATHFINDERCHALLENGES-01 HEISINGBERG Project 101114978 and the support from Weizmann-UK Make Connection Grant 142568.

**Author contributions** K.P.K., J.G., J.C., J.H.C., D.C., D.J.K. and B.R. contributed equally to this work. F.P., C.G. and H.B. conceived of the project. K.P.K., J.G. and C.G. developed and designed the abstraction and corresponding unification with the support of N.G.B. and H.B. F.P., J.H.C., D.C. and L.P. developed the hardware and designed its optical and electrical parts. J.C., J.G. and D.J.K. performed the ML experiments and analysed their data. J.C., K.P.K., J.H.C., C.G. and F.P. performed the optimization experiments and analysed their data. C.G., J.G., K.P.K., J.C. and

D.C. implemented the digital twin. J.G. designed and ran the digital twin experiments and analysed the data for ML models. K.P.K. and C.G. designed and ran the digital twin experiments and analysed the data for optimization problems. B.R. and J.G. designed and ran the generalization and noise robustness experiments. J.C., J.H.C., D.J.K., G.B., L.P., D.C. and F.P. implemented and characterized the AOC components and technologies. D.J.K., J.H.C., B.C., F.P., H.B., L.P. and D.C. developed the roadmap and the AOC calculations. D.J.K., J.H.C., J.C., G.B., L.P. and G.O'S. developed the control software. B.R., K.P.K., F.F., H.K., S.R., V.R. and H.B. provided technical input to ML experiments. M.H., K.P.K., C.G. and H.B. designed the MRI experiments. L.B., S.K., C.G., K.P.K. and H.B. designed the transaction-settlement experiments. A.R., N.G.B., S.R., V.R. and J.K. provided technical input to the hardware applications. K.P.K., J.G., H.B., J.H.C., F.P., B.R. and C.G. wrote the paper with input from all authors.

**Competing interests** The authors of the paper have filed several patents relating to the subject matter contained in this paper in the name of Microsoft Co.

**Additional information**
**Correspondence and requests for materials** should be addressed to Kirill P. Kalinin, Francesca Parmigiani or Hitesh Ballani.

a

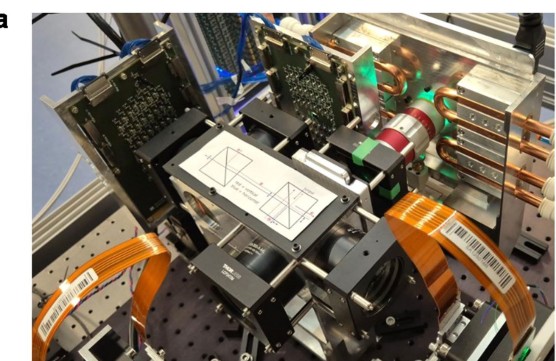

b

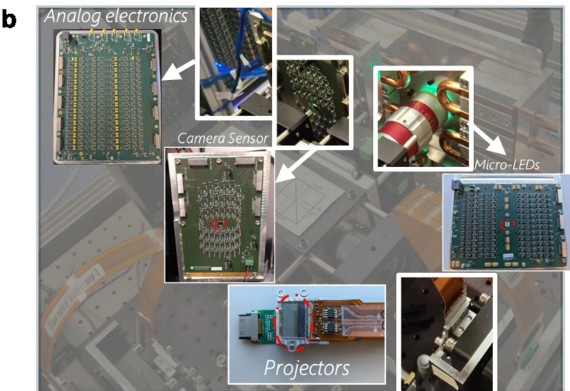

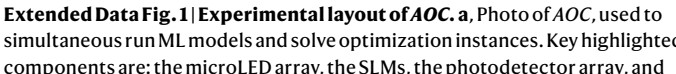

c

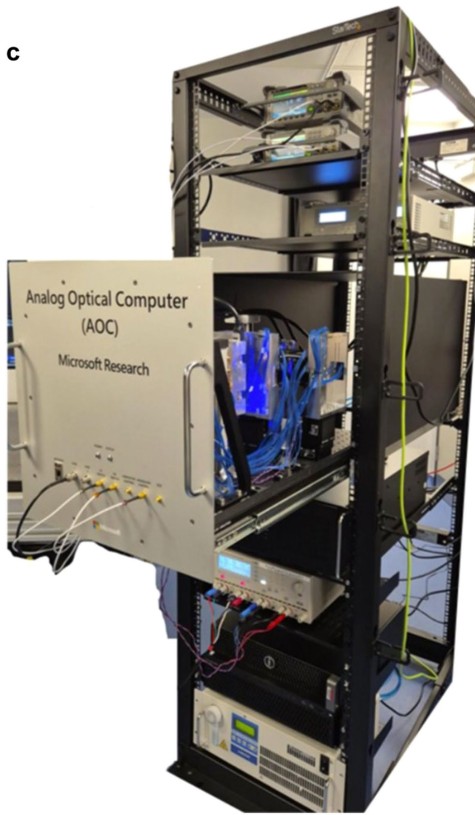

**Extended Data Fig. 1 | Experimental layout of *AOC*. a**, Photo of *AOC*, used to simultaneous run ML models and solve optimization instances. Key highlighted components are: the microLED array, the SLMs, the photodetector array, and the analog electronic block. **b**, *AOC* photo inside a rack, with all required equipment included.

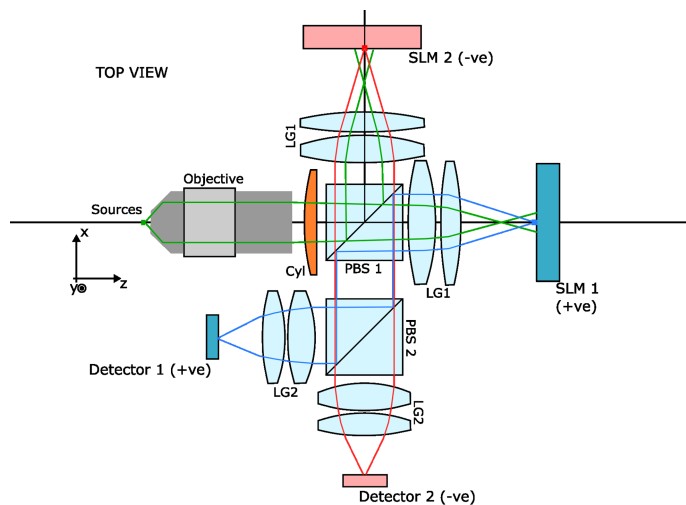

**Extended Data Fig. 2 | Detailed Schematic diagram of the optical vector-matrix multiplication in *AOC*.** Light from the sources is collected by the objective. The sources are imaged onto the SLM using the combined 4F system of the objective and lens group 1 (LG1). Polarizing beamsplitter 1 (PBS 1) splits the light by linear polarization state and sends light to one of two modulators. The reflected light is modulated by polarization (multiplication) and the action of PBS 1 makes this an amplitude modulation. Each SLM is imaged onto its corresponding detector using LG1 and LG2 through PBS 1 and PBS 2. Summation happens at the detector.

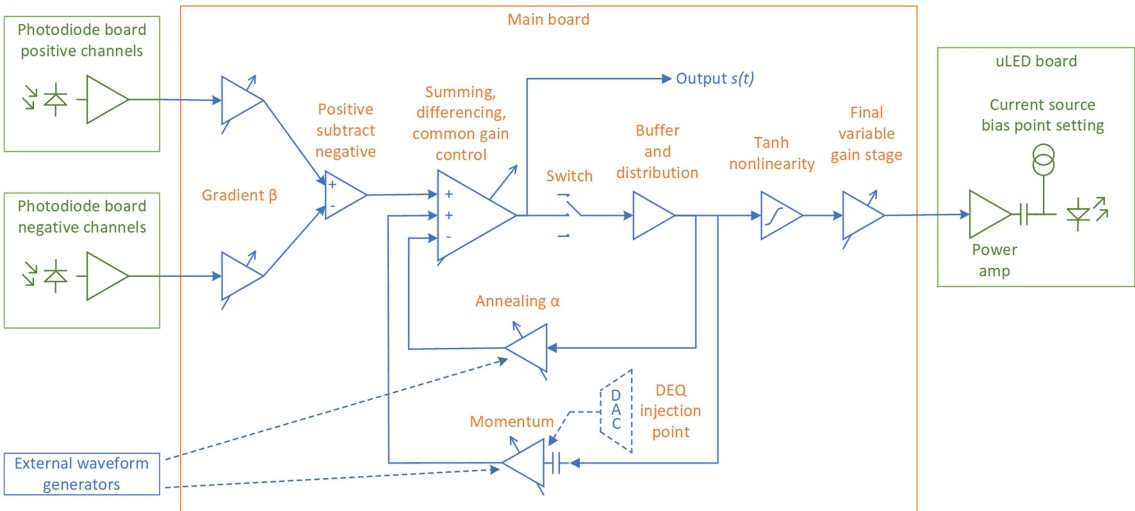

**Extended Data Fig. 3 | Schematic diagram of *AOC*.** Schematic diagram of the analog electronic parts (in the orange box) and the optical parts (in the green boxes), where the key functions and operations are highlighted. The state vector *s(t)* in equation (1) and equation (2) is measured at the pointed indicated as output.

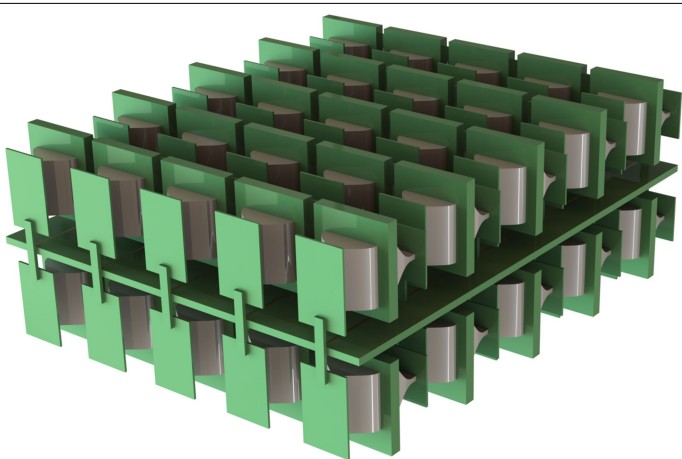

**Extended Data Fig. 4 | *AOC* at scale.** Artistic representation of *AOC* at 100 million weights scale, showing the 3D mesh structure of the required 50 modules, with each module with size of $4 \times 10^6$ weights (2000 variables) and distinct optical modules for positive and negative weights.

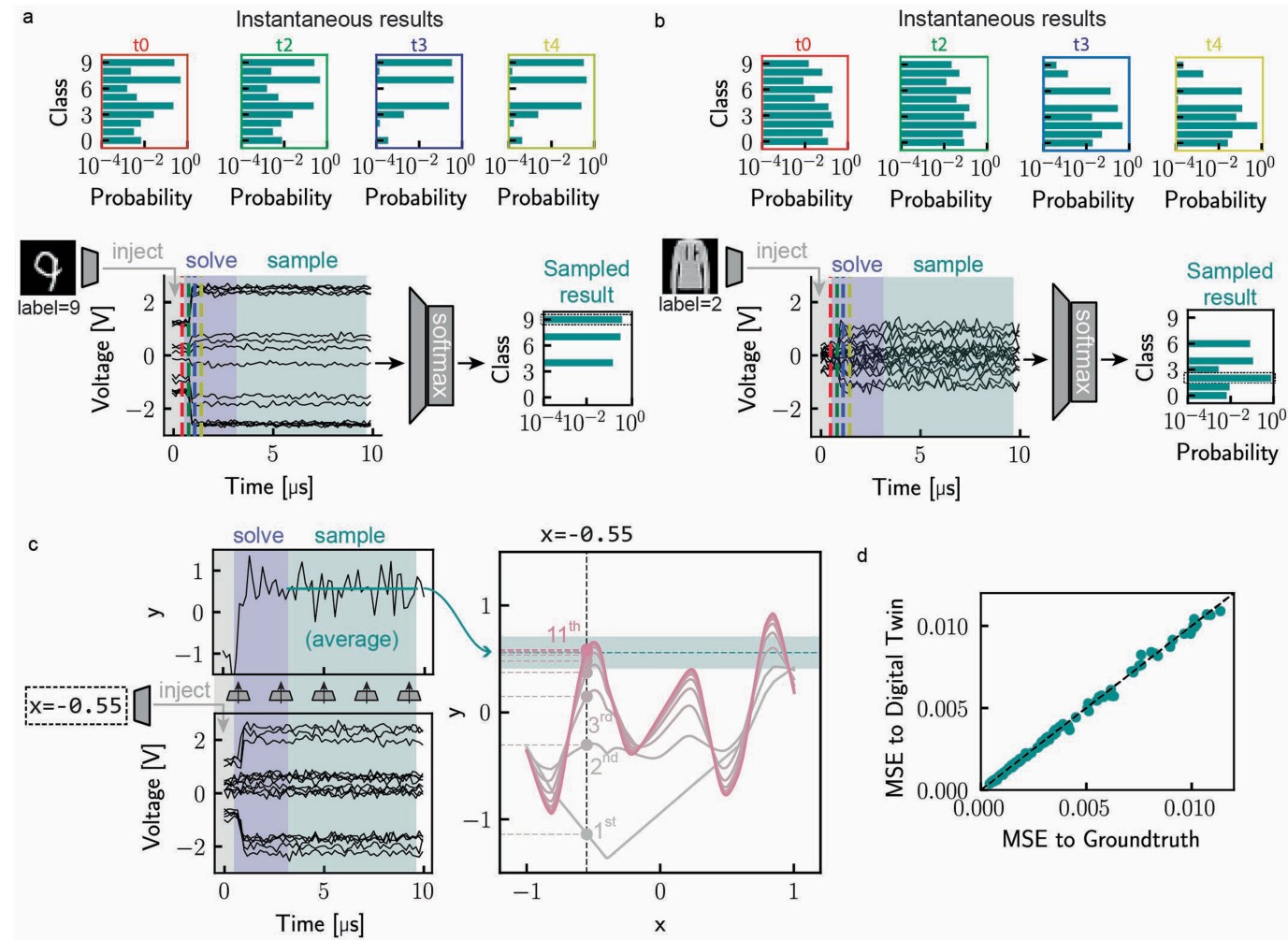

**Extended Data Fig. 5 | Traces and Multi-Seed-DT-AOC comparison.**
**a**, Example trace of an MNIST example. We selected four time points of the trace where we projected the measured state into classification space using the output projection and applied a softmax to obtain probabilities. The most likely class stays on the wrong label 7 until the fully-sampled result is evaluated in which the correct class label 9 is reached. **b**, Example trace of a Fashion MNIST example. Here, the initial entropy of the class-probability distribution is high but drastically falls off over iterations. The correct class with label 2 emerges early on and is maintained throughout the iterations and in the sampled result.

**c**, Example traces of a sinusoidal regression task for $x = -0.55$. While the trace sampled during evolution averages multiple iterations and is asynchronously sampled, we can observe some correspondence between the evolution of the *AOC-DT* for the same point. The plot on the right shows the evolution of the entire curve over *AOC-DT* iterations. The point of interest is marked by the vertical dashed line and appears to follow the same trajectory from close to –1 to its target value. **d**, Multi-seed study of the Gaussian regression task for 100 independently trained and tested equilibrium models to test how reliable the *AOC-DT* models the AOC device.

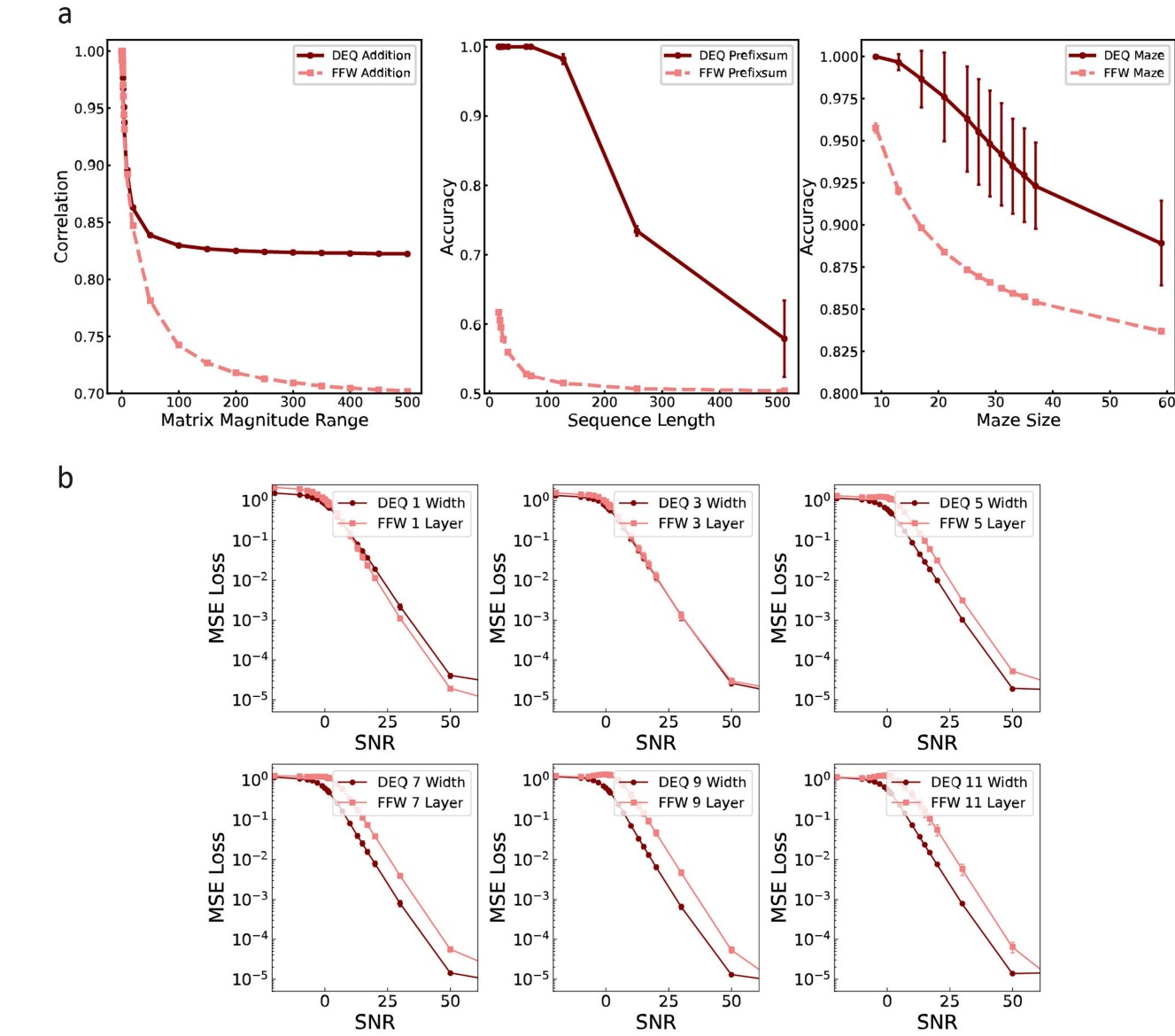

**Extended Data Fig. 6 | Increased out-of-distribution generalization and equilibrium model robustness. a**, Out-of-distribution generalization of the equilibrium model. **b**, Robustness of the equilibrium model to noise for feedforward models with varying number of layers and parameter-matched single-layer DEQs.