## [Peer Review File · Nature]

Analog Optical Computer for AI Inference and Combinatorial Optimization

Corresponding Author: Dr Hitesh Ballani

Version 1:

Reviewer comments:

Referee #1

(Remarks to the Author)

This paper presents an analog optical computer (AOC) based on a recursive architecture that integrates 3D optics and analog electronics. The key features of the AOC include: 1. Fully analog operation, eliminating the need for data converters (and their associated latency). 2. Unified memory and computation, avoiding the separation typically found in digital architectures. 3. Low latency (30 μ s), dictated by the optical path length and feedback mechanisms. One of the notable contributions of this work is the implementation of Quadratic Unconstrained Mixed Optimization (QUMO), which provides an advantage over traditional binary formulations (QUBO). The fixed-point model presented here is interesting and useful. Furthermore, the breadth of real-world problems tackled experimentally is impressive, as it moves beyond typical benchmark problems. However, the significance of these contributions does not become clear until well past the first few paragraphs. Additionally, the paper contains excessive jargon, which at times makes the work appear unnecessarily complex.

Main comments and concerns as follows:

1. The fundamental architecture (Image \rightarrow SLM \rightarrow Detectors) is not new and was previously proposed by Psaltis and Farhat (see Psaltis & Farhat, Optics Letters, 1985, and Farhat, Psaltis, Prata, & Paek, Applied Optics, 1985). I am surprised that these works are not cited. The primary distinction in this paper is the inclusion of analog electronics to close the loop, which avoids digital-to-analog (DAC) and analog-to-digital (ADC) conversions. This modification enables a 30 μ s time-to-solution, which is commendable for a non-integrated recurrent system.
2. The authors present this work as a fundamental paradigm shift; however, it may be more accurate to state that their system excels at a specific task (fixed-point search) that can be mapped onto multiple applications.
3. The paper does not discuss success probability or time-to-solution in detail, except in Figures 16 and 17 for 16-variable problems. What is the authors' stance on exponential time complexity, and how does their system compare in terms of success probability and scalability?
4. I recommend that the authors provide an analysis of time complexity, specifically:
 - Time to Solution (TTS) and Success Probability as functions of problem size (N)
 - Comparative analysis against standard benchmarks like MaxCut and the SK model
 - A comparison with existing platforms, similar to Figures 3 and 4 in DOI: 10.1038/s42254-022-00440-8, to better evaluate performance and competitiveness.
5. The author should provide a comparative table presenting accuracy, time to solution, and performance metrics relative to existing platforms would greatly enhance clarity regarding the advantages of the proposed architecture in contrast to state-of-the-art solutions.
6. The paper discusses implementing nonlinearity and the momentum algorithm within the electronic domain, which may introduce significant computational overhead and memory constraints. I suggest discussing: a) The feasibility and potential benefits of implementing these elements in the optical domain with optical feedback. b) Fundamental limitations of this approach and strategies to mitigate these challenges.

7. The authors implemented hyperbolic tangent (tanh) as the nonlinearity, but it would be beneficial to: a) Explain why tanh was chosen over other nonlinearities. b) Discuss the possibility of implementing optical nonlinearity. c) Investigate the impact of alternative nonlinearities on computational performance, as different nonlinear functions can significantly influence efficiency, even when using electronic nonlinearity~[1,2].

1. M. Miscuglio, et al. "All-optical nonlinear activation function for photonic neural networks." *Optical Materials Express* 8.12, 3851-3863 (2018).

2. F. Böhm, et al. "Order-of-magnitude differences in computational performance of analog Ising machines induced by the choice of nonlinearity." *Communications Physics* 4.1, 149 (2021).

8. The demonstrated AOC hardware consists of 16 microLEDs and 16 photodetectors, limiting it to a 16-variable state vector. While problem decomposition could extend this to 64 variables, this comes at the cost of increased computational overhead. Compared to other AOCs designed for machine learning or optimization, the scalability of this architecture is not particularly impressive.

The authors primarily discuss scalability in terms of operational speed and energy efficiency, but they should also address: a) Challenges in large-scale implementation, including synchronization, crosstalk, and error propagation. b) Potential limitations of scaling the architecture beyond current configurations.

9. In OVMM-based large-scale free-space architectures, the distance between each microLED and the SLM pixel, as well as between the SLM pixel and the detector, varies due to additional propagation paths that scale with SLM size. This results in time delays in summation at the photodetector, which could significantly impact processing speed and introduce errors. The authors should discuss strategies to mitigate these effects.

10. The authors claim a 100× improvement in TOPS/W over a GPU. However, their single unit is roughly 4U in a server rack, and their scaled system (shown in supplementary figures) includes 50 units. While trade-offs exist, expecting a 100× efficiency gain at the cost of a 100× increase in physical size seems unrealistic.

11. The authors have done thorough work in designing a functional free-space optical computing system, but free-space optics is inherently large and cannot be miniaturized significantly. Even with commercial off-the-shelf (COTS) optics, diffraction imposes a fundamental limit, making it unlikely that the system could be reduced by more than a factor of 2.

12. The authors analyze large-scale implementations toward the end of the paper, but can these systems realistically be fabricated in a compact and practical manner? The claim of two orders of magnitude improvement in power consumption warrants further scrutiny.

13. The comparisons between their system and NVIDIA GPUs feel hand-wavy. I strongly recommend that the authors provide rigorous, unbiased calculations and a scaling roadmap.

14. The authors claim 2 GHz operation, suggesting that optics, not electronics, would be the limiting factor. However, in practice, I would expect electronics to impose constraints rather than optics.

15. While this is an impressive system, the long-term vision is unclear. The introduction and abstract should be more clearly written, explicitly stating that the core contribution is a fixed-point solver using mixed linear optics and nonlinear analog electronics in a feedback loop.

Referee #2

(Remarks to the Author)

A. Summary of the key results

In this paper, the authors present a machine learning and optimization platform based on a combination of a 3D optical matrix vector multiplier based on microLEDs, spatial light modulators and photodiodes, and an analog feedback mechanism. They demonstrate inference on an overwhelming array of benchmark tasks and report impressive results. The results in the paper are certainly very impressive for this type of alternative computing platform.

B. Originality and significance

The close integration of analog electronics, optical hardware and feedback mechanism is more thoroughly done than previously, and the hardware has been tested on a much wider array of problems than previously seen.

C. Data & Methodology

There is a huge quantity of data and results presented. However, my main criticism of the paper My main criticism of the paper is that it is hard to follow what was actually done and how it was done, as they spend more time discussing the results in great detail than explaining the nuts and bolts of the implementation. I think a reorganization of the main text of the paper with more focus on implementation and only summarization of the results would be helpful. In general, I would recommend moving some of the test results into the supplementary information, and putting more information on how the platform works and how it was implemented into the main text with only a summary of results. A table of results summarizing performance, including benchmarks like time and energy as well as accuracy, and showing the results with only the input/output processing and without the hardware is recommended.

D. Appropriate use of statistics and treatment of uncertainties
The results are very thoroughly presented.

E. Conclusions, robustness, validity, reliability

Equilibrium/fixed point networks have been a subject of significant interest for some time, but have not been practically implemented in machine learning/AI. In general, they require architecture restrictions (such as symmetric weights) and do not allow implementation of the arbitrary and specific architectures that have been shown to perform so well on modern machine learning tasks. A discussion of why the authors think this will scale to perform modern machine learning tasks and why that has not been previously observed would be very helpful. In particular, how computationally expensive would performing the tasks that their hardware implements be in silico? Is this part of the reason that it is not an approach typically used in ML? Relatedly, how computationally expensive was the digital twin training?

F. Suggested improvements: experiments, data for possible revision

- It is clear that the data in the problems have input and output processing (from Fig 1., which indicates this), as well as from the size of the inputs and weight matrices (16 inputs, 256 weights), which is too small alone to solve fashionMNIST or MNIST, which have input sizes of 784 and are typically solved with networks with $> 10^5$ parameters (weights). A clear description of these processing steps, as well as reports on a control of how well the benchmark tasks do with this specific input/output processing and without the optical/analog recurrence (rather than generically how well you can do with linear solves) is required for publication. This type of sweeping under the rug is not acceptable.

- Equations 1 and 2 are extremely helpful for understanding what the authors are implementing. However, it is very difficult to understand from the paper how this is implemented in the analog processing, for example how the state variable $s(t)$ is related to a physical parameter in the microLEDs or the photodiodes, and how the operations in the schematic shown in Supplementary Fig. 7 (which should be a figure in the main text, or at least in the methods section) relate to the equations. I would like to see the relationship between the abstract equation and the physical reality.

- To implement the 4096 model, some kind of multiplexing is performed in time. How the data is broken up and how this is performed is not adequately described. A figure and description about how time and energy scales when this is done would be helpful.

G. References

References are appropriate

H. Clarity and context: lucidity of abstract/summary, appropriateness of abstract, introduction and conclusions

Abstract and introduction are acceptable. More details on implementation with less nitty gritty results descriptions in the main text would be helpful for clarity.

Referee #3

(Remarks to the Author)

The authors introduce the Analog Optical Computer (AOC), a novel platform that integrates analog electronics and three-dimensional optics to accelerate both machine learning inference and combinatorial optimization. By leveraging rapid fixed-point search, AOC eliminates the need for frequent digital conversions and enhances noise robustness. The hardware is designed to support emerging compute-bound neural models, including those with recursive reasoning and advanced gradient-based optimization methods. The manuscript presents rigorous methodology and compelling experimental validation across a diverse set of machine learning and optimization problems, including challenging synthetic QUMO and QUBO benchmarks. The case studies are well-chosen, covering applications from image processing to financial modeling, demonstrating AOC's versatility and real-world potential.

This work is original and represents a significant advancement in analog optical computing. The authors' co-design approach—aligning the hardware with the ML and optimization algorithms—positions AOC as a transformative computing architecture. The manuscript is clearly written, well-organized, and methodologically sound.

While this is a strong and highly promising work, a few refinements would strengthen the manuscript:

- A more direct comparison with emerging alternative accelerators (e.g., D-Wave Leap Hybrid Solver, Fujitsu Digital Annealer) would provide additional context on AOC's relative advantages, particularly in energy efficiency and scalability.
- While the manuscript convincingly argues that AOC can scale, additional discussion on manufacturing feasibility and cost implications would be valuable.
- A discussion on hardware limitations, error tolerance, and trade-offs in precision would provide a more comprehensive perspective on AOC's practical deployment.

Overall, this is an exceptional manuscript with strong experimental results and significant impact on energy-efficient, scalable computing. With minor clarifications and refinements, this already strong work will be even more impactful. I highly recommend publication, pending these refinements.

We thank all the reviewers for their thorough and insightful comments. Their suggestions have greatly helped improve the clarity and presentation of our key findings, and we are excited about how the updated manuscript will resonate with Nature's broad readership.

Below, we provide detailed point-by-point responses to all Reviewers' comments. Before that, we summarize the main changes:

- (1) **Manuscript readability:** we have shortened the introductory text, reinforced the connection between the AOC hardware and its underlying mathematical abstraction, and improved the overall flow of the ML section. Additional refinements throughout the manuscript enhance clarity and coherence.
- (2) **AOC scalability:** while the paper focuses on the potential of codesigning hardware and abstraction for analog computing and implications for acceleration of real-world applications, we acknowledge the challenges of scaling this technology. The current hardware implementation is indeed small-scale. We, however, clarify a potential path for scaling AOC to practical application sizes and highlight key challenges being actively explored in the research community. Furthermore, we emphasize that AOC's hardware architecture is designed to offer scalability and manufacturability advantages over many other analog and optical computing approaches.
- (3) **Benchmarking:** we have added additional benchmarks and comparisons:
 - In response to Reviewer 1's comments on optimization, we have included an additional scaling benchmark for the requested SK and MaxCut QUBO graphs, complementing the multiple benchmarks already present in our manuscript.
 - We have relocated the feedforward model results from the Supplementary Information to Figure 2, in response to Reviewer 2's feedback on the ML section, to enhance their visibility.
 - We have elevated results demonstrating the inherent error tolerance of recurrent models to Extended Data Figure 6, following Reviewer 3's comments.
 - Following Reviewer 3's suggestion, we have added a new section in the Supplementary Information discussing the quantum hardware approaches to optimization and their limitations relative to AOC.

Thanks to the Reviewers' comments, we believe these revisions have strengthened the manuscript and improved its clarity, rigor, and scope.

Reviewer 1

[1] The fundamental architecture (Image → SLM → Detectors) is not new and was previously proposed by Psaltis and Farhat (see Psaltis & Farhat, Optics Letters, 1985, and Farhat, Psaltis, Prata, & Paek, Applied Optics, 1985). I am surprised that these works are not cited. The primary distinction in this paper is the inclusion of analog electronics to close the loop, which avoids digital-to-analog (DAC) and analog-to-digital (ADC) conversions. This modification enables a 30 μs time-to-solution, which is commendable for a non-integrated recurrent system.

We are grateful for the suggestions; the updated manuscript now cites both papers. We now cite *Psaltis & Farhat, Optics Letters, 1985* as Reference [9] in updated Introduction:

“Analog approaches – leveraging optics [1-4,9], ...”

And we cite both suggested papers as References [7-8] in Supplementary, Section B:

“The Stanford optical vector by matrix multiplication design uses an interleaved 4F and 2F system with cylindrical lenses [6-8].”

in addition to *J. W. Goodman et al (1980)* that we were referencing earlier.

[2] The authors present this work as a fundamental paradigm shift; however, it may be more accurate to state that their system excels at a specific task (fixed-point search) that can be mapped onto multiple applications.

The main text of the paper is adjusted accordingly to emphasize the fixed-point nature of algorithms that our hardware is suitable for. The relevant text from the Introduction is as follows:

“We introduce the analog optical computer (AOC), the first non-traditional computing platform designed for both AI inference and combinatorial optimization. By combining optical and analog electronic components within a feedback loop, AOC rapidly performs fixed-point search without digital conversions.”

[3] The paper does not discuss success probability or time-to-solution in detail, except in Figures 16 and 17 for 16-variable problems. What is the authors’ stance on exponential time complexity, and how does their system compare in terms of success probability and scalability?

Exponential time complexity: The exponential time complexity in combinatorial optimization is fundamentally hard to break. The advanced gradient descent algorithm we propose is a heuristic algorithm with no guarantees of finding the global minima solutions in polynomial time. When implemented in AOC hardware at scale, it has the potential to offer two orders of magnitude speed advantage over its digital implementations.

Success probability: Indeed, the success probability is a standard metric that is commonly discussed for heuristic solvers, and that is why several success rate maps are included in our manuscript too (see Suppl. Figs. 13, 14, 16 in updated manuscript). At the same time, relying only on the success rate as the main metric could be misleading due to the often-overlooked effort in tuning hyperparameters. This is why our digital twin solver is designed as a black-box optimizer, incorporating parameter search within the given time budget. Besides parameter tuning, the sensitivity of success probability to hyperparameters is also critical: spending a considerable amount of time to identify hyperparameter settings that result in very high success probability may be less desirable (from an end-to-end wall-time point of view) than quickly identifying hyper-parameter settings that still result in finding the best solution, although with smaller success probability. This is why we present the number of samples metric to achieve certain quality of solution in the main part of the manuscript.

Scalability: While current hardware supports up to 16 variables (or a few tens via block-coordinate descent), our digital twin demonstrates competitive performance across various QUBO and QUMO benchmarks, the instances within which reach up to 20,000 variables, indicating the potential of the algorithmic approach behind AOC at scale.

Following the Reviewer’s feedback, we added to the metrics discussion the following (Section H.5: AOC-DT performance at scale in optimization, Suppl. Information):

“We note that relying only on the success rate as the main metric could be

misleading due to the often-overlooked effort in tuning hyperparameters. This is why our digital twin solver is designed as a black-box optimizer, incorporating parameter search within the given time budget.”

We also added references to two relevant suppl. sections with benchmarking details in the main part of the paper (Section “AOC for Optimization”):

“The details about AOC-DT parameters are provided in Supplementary Sec. H.4 with additional benchmarks in Supplementary Sec. H.5.”

[4] I recommend that the authors provide an analysis of time complexity, specifically:

Time to Solution (TTS) and Success Probability as functions of problem size (N)

Comparative analysis against standard benchmarks like MaxCut and the SK model

A comparison with existing platforms, similar to Figures 3 and 4 in DOI: 10.1038/s42254-022-00440-8, to better evaluate performance and competitiveness.

One of the key contributions of our work in optimisation is the proposition of the quadratic unconstrained mixed optimisation (QUMO) formulation with its practical applicability illustrated through applications (transaction settlement, MRI reconstruction). Nevertheless, given the historical importance of the QUBO abstraction, the original manuscript includes a comprehensive comparison of our AOC-DT solver across multiple benchmarks to the Suppl. Materials (namely Wishart, G-Set, Tile3D, etc, see Supp. Figure 17). For the reasons explained in the answer to Q3, we compare our solver as a black box solver against other heuristic approaches implemented in a similar black box solver fashion (parallel tempering, simulated annealing), as well as Gurobi.

At the same time, we appreciate the Reviewer’s comment and agree that additional QUBO scaling benchmarking could be done. Following the Reviewer’s suggestion, we have contacted the authors of the suggested paper (DOI: 10.1038/s42254-022-00440-8) and understood from them that since this paper is an overview of the state-of-the-art, each team used different SK and MaxCut instances in the benchmarking figures (eg Fig3, Fig4). Therefore, we generated similar SK and MaxCut instances ourselves. We note that their SK model doesn’t have the classic Gaussian distribution for matrix coefficients and instead has elements from $\{-1, 1\}$ domain, which is known as SK with bimodal distribution (SK-bimodal, see eg M. Aramon, *Frontiers in Physics* 7, 2019). Also, their MaxCut problem construction corresponds to the Erdős-Rényi model with edge probability 0.5.

The scaling analysis of AOC-DT for the generated SK and MaxCut instances appears now in Supp. Figure 19 (reproduced below) and in a new paragraph in section H.7 in the appendix. Supp. Figure 19 directly compares to Figures 3 and 4 in the overview paper, with the following minor differences:

- (a) since we present results only for AOC-DT (and do not need to worry about visual clutter as in the suggested overview paper), we provide box plots of the performance results that reveal more statistical information;
- (b) due to the use of box plots, the x-axis is not linear in our plots;

(c) while generating the MaxCut instances, we further enforce the condition that the generated graph must be connected (this is a natural condition relevant mostly to the very small graphs);

(d) we slightly adapted equation (9) of the overview paper to be

$T = \tau \cdot \max\left(1, \frac{\ln 0.01}{\ln(1-p_{succ})}\right)$, to account for the fact that one experiment must always complete even when the success probability is very high (again, this is relevant only to the very small problems);

(e) We report success rate metric for similar problem sizes as in the overview paper, showing the competitive performance to other platforms.

(f) We report time-to-solution metric in terms of number of iterations, rather than seconds, to draw attention that these results are based on simulations and are not measured in hardware. Multiplying it by the estimated round-trip time of the AOC hardware, one can get an estimate of the standard TTS.

The data, used to produce the scaling for AOC-DT, is provided as part of the paper and, hence, interested readers can visualize them in different ways.

The new Supp. Figure 17 is as follows:

Supplementary Fig. 17: Success probabilities (a and b) and time to obtain a 99% success probability of obtaining the ground state (TTS) (c and d) of the AOC-DT for Sherrington-Kirkpatrick (SK) (a and c) and dense MaxCut (b and d) problems, as a function of the problem size (N). Observe the logarithmic scale in the y-axis, and the non-linear x-axis. Graph construction and evaluation follows³⁶ (Figures 3 and 4). The Ising graphs used as input in a and c are constructed by creating a symmetric $N \times N$ matrix where each non-diagonal entry takes the value -1 or 1 with equal probability (diagonal entries are 0). The MaxCut graphs used as input in b and d are constructed by creating a symmetric $N \times N$ matrix where each non-diagonal entry takes the value 0 or 1 with equal probability; we further reject graphs that are not connected. For each input graph size (N) we construct 10 matrices and provide aggregate performance statistics as box plots. The TTS metrics is reported as the number of required iterations to obtain the ground state with 99% success probability; ground state is defined as the best solution found by the solver. Hence, a lower bound of the actual wall-time is the reported TTS number multiplied by the round-trip time of the system (ignoring all other system overheads).

[5] The author should provide a comparative table presenting accuracy, time to solution, and performance metrics relative to existing platforms would greatly enhance clarity regarding the advantages of the proposed architecture in contrast to state-of-the-art solutions.

We followed the Reviewer's suggestions and produced the scaling plots (focussing on time to solution and success probability) similar to Figs3-4 from the overview paper, which is an addition to the already multiple QUBO benchmarks presented in our paper: Wishart, G-Set, RCDP, Tile3D, QPLIB-QUBO, with the detailed tables of the AOC accuracy (as well as Gurobi, parallel tempering, simulated annealing) provided as part of the data release process.

[6] The paper discusses implementing nonlinearity and the momentum algorithm within the electronic domain, which may introduce significant computational overhead and memory constraints. I suggest discussing: a) The feasibility and potential benefits of implementing these elements in the optical domain with optical feedback. b) Fundamental limitations of this approach and strategies to mitigate these challenges.

Computational overhead due to nonlinearity and momentum within electronic domain:

at each iteration, our architecture efficiently transitions from optical to analog electronic domain, where nonlinearity and momentum are implemented. When going from analog electronics to optical domain, the input (x) to the matrix-vector multiplier is realized by optical sources with possible conversion efficiencies of around 30%. When going from optical domain to analog electronics, the output of the matrix-vector multiplier ($y = Wx$) is measured by Si-photodetectors with responsivity of around 40% at the operating wavelength. Staying in the analog domain throughout each iteration of the fixed-point algorithm eliminates the substantial additional energy overhead associated with repeated analog-to-digital and digital-to-analog conversions.

Memory constraints due to nonlinearity and momentum within electronic domain: Our in-memory architecture ensures that signals remain in the analog domain until convergence, eliminating the need for data movement.

- a) **Feasibility of implementing nonlinearity in optical domain:** While all-optical architectures have been explored extensively, including those using planar and 3D optics [see Refs 4-7 and 11-14 from main manuscript], most existing approaches rely on feed-forward models and ultimately resort to digital processing for nonlinearity. Optical nonlinearities do exist, for example, in the work referenced by the Reviewer (Miscuglio et al.), where it is achieved using a reverse saturable absorber. This requires an additional optical pump per nonlinear transfer function, operating with femtosecond-scale optical pulses.

Feasibility of implementing momentum in optical domain: A fully optical feedback system, like the one proposed by Marandi et al. (2014), showcases an all-optical Ising machine. A similar (but simplified) approach could be used to create momentum term at the cost of dealing with the coherent light.

- b) **Fundamental limitations of nonlinearity in optical domain:** While implementing nonlinearity in the optical domain is an extremely interesting concept and an active research area, it uses bulky, high-power sources and is incompatible with miniaturized, low-power, and low-cost 3D optical technologies amenable to volume manufacturing—key elements for achieving a scalable AOC architecture with 100x higher efficiency.

For example, the approach in M. Miscuglio et al. (2018) requires centimeter-scale nonlinear optical devices and high-energy coherent femtosecond optical pulses. The need for such short pulses significantly increases energy consumption and implementation costs, making this approach less practical for scalable, low-cost ML and optimization applications, compared to analog electronics. Furthermore, implementing nonlinearity in the analog electronic domain provides greater

flexibility, allowing us to easily experiment with different nonlinear functions for future applications.

Fundamental limitations of momentum term in optical domain: For the fully optical feedback Ising machine mentioned above, scaling was ultimately achieved by replacing optical feedback with electronic control via FPGAs. This highlights the fundamental difficulty in scaling coherent optical architectures due to the requirement of precise optical path matching at THz frequencies. Our system, by contrast, is based on incoherent optical sources, which significantly simplifies scaling. Unlike coherent systems that require optical paths to match at frequencies of 100s of THz, our approach only requires GHz-level matching.

Following the Reviewer's comments, we have now added in the main body of the manuscript (Section 6 Discussion: AOC Scalability and Efficiency of the main manuscript):

"Furthermore, since microLEDs are incoherent light sources, optical paths need to be matched only within the system bandwidth (GHz) rather than at the source wavelength (hundreds of THz), which is a fundamental manufacturability advantage over coherent systems."

And further clarified in the third paragraph of A Scaling, computing speed, and energy estimations section in Supplementary Information:

"Our architecture efficiently transitions between optical and analog electronic domains at each iteration, where nonlinearity and momentum are implemented. Optical-to-electronic conversions and vice versa are performed using Si-photodetectors and microLEDs, eliminating the energy overhead of repeated analog-to-digital and digital-to-analog conversions. The in-memory design ensures that signals remain in the analog domain until convergence, mitigating data movement constraints.

An important factor in scalability is our choice of surface-emitting sources and photodetectors, which are crucial for leveraging the third dimension, where the computation is happening. MicroLEDs and Si Photodetectors can be grown on silicon, enabling tight integration with an electronic backplane to support large arrays."

Also, we added a discussion about optical nonlinearities to Section B.2:

"Alternative ways to realize nonlinearity in optical domain.

Over the years, numerous impressive efforts have been made to achieve efficient optical nonlinearities, see reference [9] and additional citations within this paper. However, these approaches typically require high peak-power optical pumps, which are often incompatible with low-power, scalable implementations. Similarly, while all-optical loop architectures have been successfully demonstrated [10], they generally rely on coherent signals for feedback, imposing stringent precision requirements that are in the order of the signal wavelength (hundreds of THz). On the other hand, by leveraging incoherent optical sources, amenable to wafer-scale manufacturing, our architecture, based on incoherent signals, requires only GHz-level optical path matching, potentially enabling a more scalable and efficient AOC system."

Where Suppl. References [9-10] are:

[9] Y. Zhang, J. Wu, L. Jia, Y. Qu, Y. Yang, B. Jia, D. J. Moss, Graphene Oxide for Nonlinear Integrated Photonics. *Laser Photonics Rev* 2023, 17, 2200512. <https://doi.org/10.1002/Lpor.202200512>

[10] Takata, K., Marandi, A., Hamerly, R. et al. A 16-bit Coherent Ising Machine for One-Dimensional Ring and Cubic Graph Problems. Sci Rep 6, 34089 (2016). <https://doi.org/10.1038/srep34089>.

[7] The authors implemented hyperbolic tangent (tanh) as the nonlinearity, but it would be beneficial to: a) Explain why tanh was chosen over other nonlinearities. b) Discuss the possibility of implementing optical nonlinearity. c) Investigate the impact of alternative nonlinearities on computational performance, as different nonlinear functions can significantly influence efficiency, even when using electronic nonlinearity~[1,2].

a) Choice of tanh as nonlinearity:

Our analog optical computer is designed to support both machine learning inference and optimization applications. The hyperbolic tangent (tanh) function is widely used as an activation function in neural networks and is also a common nonlinearity in heuristic algorithms for solving optimization problems with binary variables. More recently, parameterized tanh is used as an alternative to normalization layers (<https://arxiv.org/abs/2503.10622>). For these reasons, tanh is a natural choice for our system as it effectively serves both domains. Additionally, tanh can be efficiently implemented in analog electronics using a bipolar differential pair, further justifying our selection.

b) Optical nonlinearity implementation:

Please see our response to Q6(b) and corresponding changes to the manuscript.

c) Impact of alternative nonlinearities on computational performance:

We agree that different nonlinear functions can influence the efficiency of the optimization solver. To investigate this, we conducted an extensive study on alternative nonlinearities. Specifically, for a subset of GSet benchmark (instances 1–21), we evaluated performance using different nonlinearities in the equation:

$$dx/dt = -\alpha f_1(x) + \beta f_2(J f_3(x)) \quad (1)$$

where f_1, f_2, f_3 were chosen from {sin, cos, tanh, linear, clipping, sign} at random. Our results are shown below:

Here the normalized performance is computed as $(\text{maxcut_found} - \text{maxcut_trivial}) / (\text{maxcut_best} - \text{maxcut_trivial})$. The x-axis methods represent combinations of

nonlinearities f1-f2-f3 with respect to the equation (1). It turned out that sin, cos, and tanh nonlinearities in place of f3, when f1 and f2 are linear, perform nearly identically with the highest relative maxcut proximity over 94%.

However, we found that adding annealing and momentum terms had a much more significant impact on performance:

- When evaluating the fraction of GSet instances (1–21) optimized to their best-known solutions, we observed that the standard Hopfield network had lower performance (denoted in legend as “Hopfield”)
- Adding annealing improved performance significantly (Hopfield+Annealing)
- Further incorporating momentum led to even better results (Hopfield+Annealing+Momentum)

This analysis indicates that the additional terms, such as annealing and momentum, are more critical to solver efficiency than the specific choice of nonlinearity. We also note that certain nonlinear functions, such as a parameterized tanh, could offer slight improvements over simpler nonlinearities such as sign() at the cost of an additional parameter for fine-tuning.

Thus, while having flexibility to implement different nonlinearities is beneficial, our study suggests that incorporating annealing and momentum terms has a far greater impact on solver performance. We have added a summary of these findings to the Supplementary Information Section H.1, stating:

“The impact of alternative nonlinearities on computational performance has been investigated for optimization. While different nonlinear functions can influence solver efficiency, we found that a small set, including tanh(), sin(), and cos(), perform similarly. More importantly, the addition of annealing and momentum terms led to substantial algorithmic improvements, significantly enhancing optimization performance.”

[8] The demonstrated AOC hardware consists of 16 microLEDs and 16 photodetectors, limiting it to a 16-variable state vector. While problem decomposition could extend this to 64 variables, this comes at the cost of increased computational overhead. Compared to other AOCs designed for machine learning or optimization, the scalability of this architecture is not particularly impressive.

The current AOC hardware is indeed limited to small-scale problems (16 variables or a 256-weight system). However, the core contribution of our work is demonstrating a single hardware platform capable of running both ML and optimization applications in fully analog manner using a feedback loop—a feature that makes the proposed AOC distinct from other architectures. While the present scale is modest, we have expanded on pathways, including challenges, in the manuscript for scaling the system to tens of thousands of variables or billions of weights. Additionally, our digital twin simulations closely match experimental results, reinforcing the algorithmic feasibility of larger-scale implementations.

[9] The authors primarily discuss scalability in terms of operational speed and energy efficiency, but they should also address: a) Challenges in large-scale implementation, including synchronization, crosstalk, and error propagation. b) Potential limitations of scaling the architecture beyond current configurations.

a) Challenges in large-scale implementation:

Synchronization: Digital twin simulations indicate that efficient performance requires module delays to remain below one iteration, corresponding to only a few centimeters of discrepancy between modules at the system’s operating bandwidth. This specification is expected to be feasible in our design. Based on the Reviewer’s feedback, we have clarified this point in the manuscript (Supplementary Information, A Scaling, Computing Speed, and Energy Estimations):

“We note the difference in delays among modules needs to be below one iteration time, to avoid the need for an analog clock to synchronize modules.”

Crosstalk: As the system scales, we plan to maintain similar size-to-pitch ratios between μ LEDs and photodetectors, ensuring that crosstalk levels remain comparable to those in our current setup. This detail has been added to the manuscript (Supplementary Information, A Scaling, Computing Speed, and Energy Estimations):

“We propose that each μ LED has an active area of 5 μ m and that the array pitch is 10 μ m, to maintain the same ratio of the current implementation.”

Error propagation: As highlighted in the manuscript, co-designing applications with unconventional hardware is key to leveraging AOC’s strengths (e.g., speed) while mitigating its limitations (e.g., noise). The fixed-point search algorithm we employ inherently provides noise robustness, ensuring reliable performance even in the presence of analog imperfections.

b) Potential limitations of scaling the architecture beyond current configurations:

To mitigate potential limitations and risks, we discuss the following key hardware considerations for scaling:

- **Use of incoherent optical sources:** This choice significantly relaxes optical path matching requirements, making scaling more feasible.
- **Selection of scalable technologies:** We utilize μ LEDs, CMOS sensors, projector displays, and analog electronics—components that are either already widely used or expected to become commodity technologies, ensuring clear pathways to low-cost, high-volume production in a compact form factor.

- **Miniaturization of the optical system:** We discuss strategies to reduce optical path lengths to the centimeter scale using micro-lenses and waveguides.

While the first two points present relatively low risk, the third requires further investigation to validate its feasibility at scale. However, there are promising advancements in scalable laser-written waveguide devices (e.g., Sotirova et al., 2024) and 3D-printed micro-optics (e.g., Weber et al., 2024), which indicate a viable path forward. Following the Reviewer’s comment, we have updated the second paragraph in Supplementary Information, A Scaling, Computing Speed, and Energy Estimations as following:

“Optical and electronic integration is essential to the scaling effort. On the optical side, the scaling will require miniaturising current 3D optics through arrays of micro-lenses, possibly in combination with 3D waveguides for fan-in and fan-out, aiming for an overall path length of a few centimeters per module. While further investigation is needed to confirm such design feasibility at scale, recent advancements in laser-written waveguide devices [1] and high-precision 3D printing via two-photon polymerization [2] provide promising directions.”

Where [1-2] are Supplementary references:

[1] Sotirova, A.S., Sun, B., Leppard, J.D. et al. Low cross-talk optical addressing of trapped-ion qubits using a novel integrated photonic chip. *Light Sci Appl* 13, 199 (2024). <https://doi.org/10.1038/s41377-024-01542-x>.

[2] K Weber, S Thiele, M Hentschel, A Herkommer, H Giessen, Positional Accuracy of 3D Printed Quantum Emitter Fiber Couplers, *Advanced Quantum Technologies* 7 (11), 2400135.

[10] In OVMM-based large-scale free-space architectures, the distance between each microLED and the SLM pixel, as well as between the SLM pixel and the detector, varies due to additional propagation paths that scale with SLM size. This results in time delays in summation at the photodetector, which could significantly impact processing speed and introduce errors. The authors should discuss strategies to mitigate these effects.

It is certainly true that the distance between each microLED and its image on the SLM, and between the SLM weights and the detectors, varies with distance from the optical axis and therefore scales with SLM size. To mitigate this effect, we employ a 4F imaging system for both the source-to-SLM and the SLM-to-detector optical paths (for more details, please see section B “AOC hardware implementation” in the Suppl. Mat.). In such 4F-like optical systems, the imaging operation equalises the path length to within the aberration of the optical system, typically a few optical wavelengths. This approach significantly minimizes the impact of propagation-induced time delays, preserving processing accuracy and speed. We added a sentence to explain this in Section B.1 Optical subsystem of matrix-vector multiplication of the Supplementary Information:

“There is an additional benefit to our 4F-like architecture, which is that in such optical systems, the imaging operation equalises the path length to within the aberration of the optical system, typically a few optical wavelengths. This approach significantly minimizes the impact of propagation-induced time delays for different optical paths, preserving processing accuracy and speed.”

[11] The authors claim a 100× improvement in TOPS/W over a GPU. However, their single unit is roughly 4U in a server rack, and their scaled system (shown in supplementary figures) includes 50 units. While trade-offs exist, expecting a 100× efficiency gain at the cost of a 100× increase in physical size seems unrealistic.

[12] The authors have done thorough work in designing a functional free-space optical computing system, but free-space optics is inherently large and cannot be miniaturized significantly. Even with commercial off-the-shelf (COTS) optics, diffraction imposes a fundamental limit, making it unlikely that the system could be reduced by more than a factor of 2.

[13] The authors analyze large-scale implementations toward the end of the paper, but can these systems realistically be fabricated in a compact and practical manner? The claim of two orders of magnitude improvement in power consumption warrants further scrutiny.

We have grouped these three questions as they are all about the challenges of scaling the proposed AOC hardware.

As the Reviewer correctly pointed out, the current AOC implementation is a free-space system housed within a 4U server rack (see Extended Figure 1). However, in a scaled version, the AOC will consist of miniaturized modules, each only a few centimeters in length, enabling support for 4 million weights (2000 variables) per module. A full system with 50-1000 such modules, with positive and negative weights separately, would scale to approximately 0.1-2 billion weights, while still fitting within a few rack units. As discussed in the main text, such approach has the potential to achieve two-order-of-magnitude improvement in power consumption.

We agree with the reviewer that achieving this level of miniaturization is one of the critical factors for scaling. While this is an ambitious goal, promising developments in scalable laser-written waveguide devices and 3D-printed micro-optics that could enable such integrated optical systems, as we noted in our answer for Q27.b, providing a research pathway toward practical optical computing platforms. Following the Reviewer's comments, we have updated the Discussion section of the manuscript as:

“Achieving the required miniaturization, however, is both a challenge and an opportunity to drive advancements in 3D optical technologies with broader applications.

While the advantage of using incoherent light over coherent sources is already discussed in Q6 and incorporated in the main text.

[14] The comparisons between their system and NVIDIA GPUs feel hand-wavy. I strongly recommend that the authors provide rigorous, unbiased calculations and a scaling roadmap.

As the Reviewer rightly points out, comparing mainstream general-purpose accelerators like NVIDIA GPUs with emerging non-traditional computing architectures is inherently challenging. In line with other work, [RefQ14-1, RefQ14-2], we computed the number of operations per second, the power consumption, and their ratio to give the headline TOPS/W figure (see Suppl. Mat. Section A “Scaling, computing speed, and energy estimations”). Computing speed can be calculated in a straightforward manner, but accurately accounting for all power consumption contributors is more complex. However, to the best of our knowledge, we have incorporated all key factors, including:

- Drivers for optoelectronic devices
- Analog-to-digital (ADC) and digital-to-analog (DAC) conversion

- Memory read/write operations

We note that since the projector refresh rate is much lower than the AOC's operating bandwidth, its power contribution remains negligible. As the system scales, these calculations can be refined and key assumptions may be revisited, such as fan-in and fan-out ratios or system power losses, ensuring that our estimates remain realistic and validated through further experimental and theoretical analysis.

[RefQ14-1] Chen, Y., Nazhamaiti, M., Xu, H. et al. All-analog photoelectronic chip for high-speed vision tasks. *Nature* 623, 48–57 (2023). <https://doi.org/10.1038/s41586-023-06558-8> - this is also reference 6 of our paper.

[RefQ14-2] M Anderson, SY Ma, T Wang, L Wright, P McMahon, Optical transformers, *Transactions on Machine Learning Research*, 2023.

[15] The authors claim 2 GHz operation, suggesting that optics, not electronics, would be the limiting factor. However, in practice, I would expect electronics to impose constraints rather than optics.

The scaled system is expected to operate at 2 GHz, which falls well within the capabilities of analog electronics, commonly used in high-bandwidth communication applications that exceed this frequency. To clarify, while some optoelectronic components – such as μ LEDs – typically operate at lower frequencies (a few kHz), their physical size can be only few μ m, reaching GHz speeds, when operating at high current levels. We recognize that referring to these solely as optical components in the manuscript may have led to some confusion. To address this, we have updated Section 6: Discussion – AOC Scalability and Efficiency to explicitly state:

“The speed is limited by the bandwidth of the opto-electronic components, which can operate at 2 GHz or higher [49].”

Where we reference:

[49] B. Pezeshki, A. Tselikov, R. Kalman and C. Danesh, "Wide and parallel LED-based optical links using multi-core fiber for chip-to-chip communications," 2021 Optical Fiber Communications Conference and Exhibition (OFC), San Francisco, CA, USA, 2021, pp. 1-3.

[16] While this is an impressive system, the long-term vision is unclear. The introduction and abstract should be more clearly written, explicitly stating that the core contribution is a fixed-point solver using mixed linear optics and nonlinear analog electronics in a feedback loop.

We hope that the long-term vision of AOC hardware has been further clarified through detailed responses to the Reviewer's questions above and multiple revisions to both the main text and Supplementary Information of the manuscript. Both, the abstract and introduction are now significantly rewritten to improve the readability, as well as shortened. To make the core contribution more explicit, the introduction now clearly states that our system is a fixed-point solver utilizing mixed linear optics and nonlinear analog electronics in a feedback loop:

By combining optical and analog electronic components within a feedback loop, AOC rapidly performs fixed-point search without digital conversions.

These modifications aim to enhance clarity for both the Reviewer and future Readers in understanding the broader impact and direction of our work.

Reviewer 2

[17] In this paper, the authors present a machine learning and optimization platform based on a combination of a 3D optical matrix vector multiplier based on microLEDs, spatial light modulators and photodiodes, and an analog feedback mechanism. They demonstrate inference on an overwhelming array of benchmark tasks and report impressive results. The results in the paper are certainly very impressive for this type of alternative computing platform.

[18] The close integration of analog electronics, optical hardware and feedback mechanism is more thoroughly done than previously, and the hardware has been tested on a much wider array of problems than previously seen.

We sincerely appreciate the Reviewer's positive assessment of our work.

[19a] There is a huge quantity of data and results presented. However, my main criticism of the paper is that it is hard to follow what was actually done and how it was done, as they spend more time discussing the results in great detail than explaining the nuts and bolts of the implementation. I think a reorganization of the main text of the paper with more focus on implementation and only summarization of the results would be helpful. In general, I would recommend moving some of the test results into the Supplementary Information, and putting more information on how the platform works and how it was implemented into the main text with only a summary of results. A table of results summarizing performance [...]

Indeed, the main text primarily focused on discussing the results, while workflow details were covered in Methods and Supplementary Information. We recognize that the workflow itself is a key result that should be carefully described in the main text. To address this, we updated the main part of the paper as follows.

We have reorganized the ML section to emphasize the workflow details. As requested by the Reviewer, numerical results for regression and classification benchmarks are summarised in a table and moved to Extended Figures. To ensure key results are still easily visible to the Readers, Figure 2d is updated and shows classification accuracies explicitly. The main text itself now describes in greater detail how digitally trained ML models are loaded into the AOC for inference. Examples of the revised main text are shown below:

“As illustrated in Fig. 2a, the complete neural network architecture includes an input projection (IP) layer, the equilibrium model, and an output projection (OP) layer. The network training is performed digitally using the AOC-DT, while the equilibrium model is deployed on the AOC hardware for inference.

[...]

During inference, the original data x goes through the IP layer as $x_{proj} = W_{IP} * x + b_{IP}$, where W_{IP} and b_{IP} are the trained IP weights and biases. For the given x_{proj} , the equilibrium model iterates on the AOC hardware until convergence, with the fixed-point state s^* read out as voltages (see Supplementary Fig. 9). Finally, the inference result is obtained by applying the OP layer to the AOC solution as $y = W_{OP}s^* + b_{OP}$, where W_{OP} and b_{OP} are the trained OP weights and bias.”

For optimization Section, the discussion was already focused on the workflow around solving several optimization problem, so we have done minor revisions in main text.

[19b] [...] including benchmarks like time and energy as well as accuracy, [...]

Time: As we mention in the Introduction, a single iteration in the current AOC takes around 20 ns. For inference tasks, convergence to fixed-points typically requires approximately 9 iterations (as estimated for AOC-DT), leading to a total inference time of 180 ns. For optimization problems, a fixed-point solution is generally reached within 30–40 microseconds, as shown in Figure 3e.

Energy: The current AOC design uses large discrete electronic components and benchtop laboratory hardware which dominates the power consumption. We believe that the energy consumption estimations, provided in Supplementary Information Section “Scaling, computing speed, and energy estimations” account for all of the fundamental energy limitations of AOC at scale.

[19c] [...] and showing the results with only the input/output processing and without the hardware is recommended.

The reviewer is asking for results obtained with “only the input and output processing” and “without hardware”. This is the purely linear regression/classification result that is indicated by the dashed line in Figure 2d. For regression, we refrained from plotting a purely linear regression line across the visibly non-linear curves in Figure 2c since this would clutter the plot and obstruct the obtained curves.

The main text of the paper is updated accordingly to make it easier to Readers to grasp these differences between the neural network architectures:

“The results above demonstrate the viability of digital training with subsequent weight transfer for opto-electronic analog inference. The contribution of the equilibrium model running on the AOC further becomes apparent when comparing the AOC results to a linear classifier, which consists of digitally trained IP, middle layer, and OP. We also train a simple feedforward model comprising IP, middle layer with tanh nonlinearity, and OP. Both linear classifier and feedforward models have the same number of parameters as the AOC.”

[20a] Equilibrium/fixed point networks have been a subject of significant interest for some time, but have not been practically implemented in machine learning/AI. In general, they require architecture restrictions (such as symmetric weights) and do not allow implementation of the arbitrary and specific architectures that have been shown to perform so well on modern machine learning tasks.

We agree that fixed-point networks appear in various forms within the machine-learning literature, including energy-based models and deep-equilibrium networks (DEQs) [17]. The networks examined in our work primarily align with the DEQ framework. DEQs do **not** require symmetric weight matrices or impose strict model architecture constraints beyond using blocks of operations with matching input and output dimensions – a condition generally met by layers in language and vision modes, where a layer is considered to be a full recurrent unit, such as a transformer block. Consequently, DEQ and self-recurrent models can, in principle, support any data modality. As noted in the main text in Section 4, both vision as well as language models have been successfully implemented within this framework:

“AOC supports neural equilibrium models, which have been widely applied across various domains from language [17] to vision [29]. [...] Their dynamic depth enables recursive reasoning, leads to improved scaling laws [32], and enhances out-of-distribution generalization compared to feedforward models [18, 33].”

Note: The above References [17, 18, 29, 28, 32, 33] are coming from the main part of our updated paper. We also highlight the absence of symmetry requirements explicitly in main text:

“In the current hardware, the equilibrium model is implemented for a single-175 layer network with 256 weight matrix, without symmetry constraints.”

[20b] A discussion of why the authors think this will scale to perform modern machine learning tasks and why that has not been previously observed would be very helpful

The updated manuscript references two very recent additional papers highlighting emerging trends around recurrent models in the machine learning community. The work [34] scales a latent recurrent model to 3.5B parameter and reports strong results on mathematical reasoning benchmarks (e.g. GSM8k). In the work [35], DEQ language models are scaled up to 1.3B parameters, demonstrating that they attain a fundamentally higher computational complexity class and can solve problems that a standard language models are provably unable to. The text in Section 4 of the main part of the paper is updated accordingly:

Recent self-recurrent language models with billions of parameters exhibit impressive reasoning capabilities, surpassing fixed-depth models in representation power [34, 35].

Where References are:

[34] J. Geiping et al. “Scaling up Test-Time Compute with Latent Reasoning: A828 Recurrent Depth Approach” (2025). arXiv: 2502.05171.829

[35] M. Schöne et al. “Implicit Language Models are RNNs: Balancing Parallelization and Expressivity” (2025). arXiv: 2502.07827.831

We note that the work in [35] is done by the members of our team.

[20c] In particular, how computationally expensive would performing the tasks that their hardware implements be in silico? Is this part of the reason that it is not an approach typically used in ML? Relatedly, how computationally expensive was the digital twin training?

As noted in Section 4, AOC-DT converges in about 9 iterations across classification and regression benchmarks. This number approximately corresponds to the theoretical slowdown of running such recurrent model on digital hardware compared to an equivalent feedforward model. In practice, time measurements of small matrix multiplications, such as 16x16 matrices, are heavily affected by kernel-launching and other fixed overheads. Additionally, by applying the implicit function theorem to compute parameter gradients at the fixed-point, we avoid storing intermediate activations. As a result, we measure a 7.84x slowdown between training-time network evaluations of a 256-parameter AOC-DT (19.6 ms) and ideal FFW (2.5 ms) models per batch of 8 on an NVIDIA A100. However, actual matrix multiplication times should be in the microsecond range due to fixed overheads mentioned above. While AOC-DT requires additional operations to model various non-idealities of our hardware, the computational cost for larger models should ultimately be dominated by matrix multiplications. Given the complexities and caveats of this discussion, we have opted to mention the 9x slowdown in the main text.

As noticed by the Reviewer, such increased computational cost of training and inference for DEQ-based models, compared to standard feedforward models, is likely a primary reason for their limited usage. We hope that emerging hardware architectures, such as the AOC, will provide efficiency gains and will spur further research into these models, making their

computational overheads more palatable. In the long term, if equilibrium models can be efficiently executed at inference, state-of-the-art generative language and image models would benefit from enhanced representational power, thanks to the increased computational complexity and test-time compute enabled by the fixed-point abstraction (see. Refs in Q20b).

Following this Reviewer’s comment, we rephrased the relevant discussion in Section 4 of the main part of the manuscript as:

“We note that due to the iterative process, the training and inference times of the AOC-DT running in silico are approximately 9× slower than an equivalent feedforward model.”

[21] It is clear that the data in the problems have input and output processing (from Fig 1., which indicates this), as well as from the size of the inputs and weight matrices (16 inputs, 256 weights), which is too small alone to solve fashionMNIST or MNIST, which have input sizes of 784 and are typically solved with networks with > 10⁵ parameters (weights). A clear description of these processing steps, as well as reports on a control of how well the benchmark tasks do with this specific input/output processing and without the optical/analog recurrence (rather than generically how well you can do with linear solves) is required for publication. This type of sweeping under the rug is not acceptable.

We acknowledge the importance of providing sufficient details to ensure Readers can easily follow the neural network architecture, including input-output processing steps. The original Figure 2(d) in the main text included the linear model baseline (dashed line):

while the comparison to the standard feedforward model was presented in Supplementary Information Section C (Figure S8, reproduced below):

It was never our intention to omit relevant benchmarking results, and we agree with the Reviewer that including this comparison in the main text adds value. The updated Figure 2(d) now incorporates the feedforward model results:

Additionally, all workflow details and processing steps are now discussed in the main text (see our response to Q37a). We note that the linear model baseline (dashed line) is obtained by training the input projection matrix (784×16), middle layer (16×16), and output projection matrix (16×10) without nonlinearities. This forms the architecture “input \rightarrow IP \rightarrow middle layer \rightarrow OP \rightarrow output,” trained and tested in the digital domain. This serves as a natural baseline illustrating the effect of bypassing the AOC. Similarly, the feedforward model (FFW), with the architecture “input \rightarrow IP \rightarrow middle layer + tanh nonlinearity \rightarrow OP \rightarrow output,” is also trained and tested digitally. Both models are parameter-matched to AOC-DT.

The key takeaway from the AOC hardware running small-scale benchmarks, such as regression and MNIST is that it demonstrated the feasibility of digital training with subsequent weight transfer for opto-electronic analog inference. The performance of AOC is comparable to parameter-matched digital neural networks, but the benefits of recurrent architectures are expected to emerge at larger scales. In addition to changes described in [Q19c], we’ve also added to Section 4:

While AOC achieves slightly higher accuracy (see Fig. 2d), the simple nature of MNIST and FashionMNIST datasets is unlikely to demonstrate the full potential of self-recurrent models. Looking ahead, this potential may materialize in a form of test-time compute in sequence modelling tasks [34,35] or inference of generative diffusion models.

Where references are the same as in our answer to [Q20b].

[22] Equations 1 and 2 are extremely helpful for understanding what the authors are implementing. However, it is very difficult to understand from the paper how this is implemented in the analog processing, for example how the state variable $s(t)$ is related to a physical parameter in the microLEDs or the photodiodes, and how the operations in the schematic shown in

Supplementary Fig. 7 (which should be a figure in the main text, or at least in the methods section) relate to the equations. I would like to see the relationship between the abstract equation and the physical reality.

a) Relationship between state variable $s(t)$ (abstract equation) and physical reality

Following the Reviewer's comment, we first simplified the description of Eq. 1 in "AOC: Fixed-point abstraction" Section (please see the updated text). In addition to the existing references to the system state equation in the "AOC: hardware" Section, including

"...in the optical domain, where the state vector s_t is encoded in the light intensity of arrays of micro light emitting diodes ... in the electrical domain using a photodetector array, where the state vector s_t is represented as a voltage per detector."

we now explicitly emphasize the dual nature of system state s_t , depending on whether it is in the optical or analog electronic domain:

In each fixed-point iteration, the analog signal alternates between optical and electrical domains, giving the system state s_t a dual opto-electronic nature.

We have also further rephrased "AOC: Hardware" section to improve readability. In addition, the circuit layout (Fig. 7 in Suppl. Inf., which is an Extended Figure in the updated manuscript) now explicitly shows where the gains α and β are set, as well as where the voltages s_t are measured in the analog electronics. The main text in "AOC: Hardware" Section is also updated accordingly:

"The circuit layout, with highlighted voltage readout position, is detailed in Extended Data Fig. 3."

We hope these clarifications would make it easier for Readers to grasp the connections between the fixed-point Eq. 1 and its realization in hardware.

b) Elevation of the circuitry figure to the main text

We have elevated it to extended figures, so it appears in the main pdf online. Given the tight figure space, we have not elevated to a figure to the main text, unless the editor would like us to do it.

[23] To implement the 4096 model, some kind of multiplexing is performed in time. How the data is broken up and how this is performed is not adequately described. A figure and description about how time and energy scales when this is done would be helpful.

We agree that our our discussion around 4096 equilibrium models may not have been clear, even though the Methods section provides most of the details the Reviewer asks (see paragraph "4096-weight ensemble model" in the Methods section). As time-multiplexing approach increases time and energy requirements linearly by the amount of multiplexing, the larger input and output projections incur only linearly more compute (i.e. matrix-vector multiplications scale linearly if only one dimension of the matrix is scaled). We updated the main text to make this information more prominent:

"In practice, model sizes tend to exceed what a given hardware can support, including traditional GPUs. To address this, we demonstrate time-multiplexing on the AOC by training a 4096-weight ensemble equilibrium model, composed of 16 independent 256-weight equilibrium models. The overall architecture mirrors the

previous 256-weight model, but the middle layer now consists of 16 independent equilibrium models, executed sequentially on the AOC for each slice of the input $x_{\text{proj}} \in \mathbb{R}^{4096}$. Classification accuracies for these time-multiplexed models are shown in Fig. 2d. The time-to-solution increases linearly with the number of independent blocks in the ensemble. Across all architectures considered for classification tasks, the IP layer accounts for the majority of parameters, while nonlinearities within the AOC primarily drive performance differences compared to a linear classifier. Additional classification results, including ablation studies of the optical and electronic contributions with untrained IP layers, are provided in Supplementary Sec. C.1.”

Furthermore, we reformulated the respective part of the Methods section:

“We can expand the model sizes supported by the hardware by using an ensemble of small models that fit on it. These smaller 256-weight models are independent at inference time but were trained jointly by receiving slices 16-sized slices of a larger input vector and stacking their outputs before the output projection. To scale to a 4096-weight equilibrium model, we expand the input space from 16 to $16 \times 16 = 4096$ dimensions and the output space from 10 to $10 \times 16 = 160$ dimensions. The input projection matrix is consequently a 784×4096 -shaped matrix and the output projection matrix is shaped 160×10 . MNIST or FashionMNIST images are scaled to the range $[-1, 1]$ and, projected to 4096 dimensions and split into 16 slices of 16 dimensions. Each of the 16 equilibrium models then runs its respective slice of input vectors to a fixed-point. Once all 16 models are run on AOC, we concatenate outputs and project them into the 10-dimensional output space where the largest dimension determines the predicted cipher.”

[24] Abstract and introduction are acceptable. More details on implementation with less nitty gritty results descriptions in the main text would be helpful for clarity.

We hope that the wide-ranging changes listed in our response to Reviewer’s questions address this one too.

Reviewer 3

We are grateful to the Reviewer 3 for their positive assessment of our manuscript.

[25] A more direct comparison with emerging alternative accelerators (e.g., D-Wave Leap Hybrid Solver, Fujitsu Digital Annealer) would provide additional context on AOC's relative advantages, particularly in energy efficiency and scalability.

AOC's unifying fixed-point abstraction: AOC's key contribution is bridging the gap between real-world applications and non-traditional hardware for both optimization and AI inference. In contrast, D-Wave's Leap Hybrid Solver and Fujitsu's Digital Annealer are prominent alternative computing platforms focused solely on optimization.

AOC QUMO abstraction and all-to-all connectivity advantage over D-Wave: Most alternative computing platforms for optimization, including the D-Wave quantum annealer, implement the QUBO abstraction in hardware, which introduces significant overhead when handling inequality constraints or continuous variables. As detailed in Supplementary H.8, mapping continuous variables to QUBO requires 10-100 times more physical variables, impeding scalability.

D-Wave's hardware faces an additional challenge: limited connectivity between physical variables. For example, in the D-Wave 2000Q system, a machine with 2048 physical variables can only support a 65-variable problem with full connectivity due to the Chimera graph topology (D-Wave QPU Solvers). Even with the improved Pegasus topology, this overhead remains more than ten times. The overhead worsens when continuous variables or inequality constraints are involved, further limiting scalability and energy efficiency. In contrast, AOC's QUMO abstraction and all-to-all connectivity avoid these inefficiencies.

D-Wave's Leap hybrid solver aims to mitigate these limitations by using a digital solver that breaks large problems into sub-components solved separately by the D-Wave hardware before recombining results. However, for the same number of physical variables, AOC can solve full connectivity problems that are 10-1000 times larger than D-Wave's hardware, depending on the number of continuous variables in a problem. While we are not aware of public numbers about the energy efficiency of D-Wave's hybrid solver, the increased requirements for the number of physical variables may indicate a significant energy penalty.

AOC vs. Fujitsu's Digital Annealer: Fujitsu's latest digital annealer (Fujitsu DAU) extends its interface beyond QUBO to support linear inequality constraints, which are common in real-world problems and motivated AOC's QUMO abstraction. However, AOC natively implements the QUMO abstraction in hardware, allowing for continuous variables, whereas handling such variables on Fujitsu's annealer would impose 10-100 times problem size overhead, depending on precision of the variables. We are also happy to see that they have confirmed several new best solutions for QPLIB benchmarks problems that AOC-DT was able to find back in 2023, in addition to finding new best solutions to other problems too (*H. Kameyama, et al. "Benchmarks for Digital Annealer with Quadratic Constrained Binary Optimization Problems", 2024*).

Public energy efficiency data for Fujitsu's digital hardware is also unavailable, but even specialized digital accelerators for AI inference, leveraging in-memory computing, typically achieve only tens of TOPS/Watt. In contrast, our energy efficiency analysis suggests that AOC's hardware can unlock significantly greater efficiency gains.

We appreciate the Reviewer's suggestion and have incorporated this comparative analysis in the new Section H.9 in Suppl. Information, highlighting AOC's further scaling and energy

efficiency advantages over the D-Wave solver for problems with dense connectivity. We also decided to add a small section there about the overall quantum hardware approach for solving optimisation problems:

H.9 Quantum hardware limitations for optimization problems

Quantum computing has emerged as a candidate hardware solver for hard optimization problems [39]. Similar to the other physical machines, quantum computers also target the QUBO abstraction. In addition to sharing the same abstraction limitations, additional quantum hardware constraints further limit the potential of quantum computing as efficient solver for optimization problems. Quantum approximate optimization algorithm (QAOA), that targets quantum gate computers, performs similar to random guess on small-size QUBO problems [39] with theoretical estimations that even a million physical qubit hardware will be still many orders of magnitude slower than existing classical heuristics [40]. The quantum computers can offer up to quadratic speed-up over classical alternatives in solving NP-hard optimisation problems, to which QUBO and QUMO belong. But, this quadratic speed-up further suffers from the slow time operation of quantum gates. Hence, unless new quantum optimization algorithms emerge and quantum computers scale significantly, it is unlikely that quantum computing will allow us to tackle challenging optimization problems at sizes of interest.

Quantum annealing platforms offer another approach for solving QUBO problems. The D-Wave pioneered QUBO hardware solvers and managed to scale from tens of variables to the current several thousands of variables over two decades [41]. In practice, one of the main challenges for this hardware is the limited connectivity of only 15 connections per each physical variable, i.e. the Pegasus topology. This translates to additional mapping overhead of QUBO problem with an arbitrary topology to the D-Wave machine. In the worst case of the fully-connected graph, the latest D-Wave Advantage hardware with 5000 qubits can accommodate only problem sizes up to 150 variables, significantly limiting scalability and energy efficiency. This one order of magnitude mapping overhead is further amplified by one-two orders of magnitude mapping advantage of QUMO over QUBO abstraction. D-Wave's Leap hybrid solver mitigates the limitations of the underlying quantum annealing hardware by coupling it with a digital solver that breaks large problems into sub-problems solved separately on hardware before recombining results digitally. However, for problems with dense connectivity, linear inequality constraints or continuous variables, the QUBO-limited nature of the hardware still impose significant scaling overheads.

Where References [38-40] are:

[39] M. P. Harrigan et al. "Quantum approximate optimization of non-planar graph problems on a planar superconducting processor". *Nature Physics* 17.3 (2021).

[40] Y. R. Sanders et al. "Compilation of fault-tolerant quantum heuristics for combinatorial optimization". *PRX Quantum* 1.2 (2020).

[41] D-Wave machine. 2023. url: <https://www.dwavesys.com/>.

In addition, the Section H.8 in Suppl. Inf. now further explains the advantages of QUMO over QUBO:

The limitations of QUBO for real-world applications have led some optimization platforms to expand their capabilities. Fujitsu's latest digital annealer [38], for example, allows linear inequality constraints to be specified in its interface. However, the underlying hardware is still a QUBO solver and even at the interface level, continuous variables still require binarization, introducing additional overhead.

Where References are:

[38] Fujitsu Digital Annealer. 2021. URL: https://www.fujitsu.com/global/documents/about/research/techintro/3rd-g-da_en.pdf

[26] While the manuscript convincingly argues that AOC can scale, additional discussion on manufacturing feasibility and cost implications would be valuable.

The manufacturing feasibility and cost implications of AOC at scale are indeed important considerations. On feasibility, as noted in our response to other Reviewers (please see Q24, Q27 and Q31), AOC’s design explicitly takes account for large-scale implementation challenges to aid manufacturing at scale and at volume. Specifically, our system is based on incoherent optical sources, which significantly eases manufacturing tolerances. Unlike coherent systems that require optical paths to match at frequencies of hundreds of THz, our approach only requires GHz-level matching. On manufacturing side, we have further clarified in the manuscript strategies to reduce optical path lengths to centimeter scale using micro-lenses and waveguides, for example, using scalable laser-written waveguide devices, and 3D-printed micro-optics, which provide viable pathways forward.

Following the Reviewer’s feedback, in addition to changes of the manuscript described in Q6, Q9, Q13, we have updated the Discussion section of the main text as follows:

“Furthermore, since microLEDs are incoherent light sources, optical paths need to be matched only within the system bandwidth (GHz) rather than the source wavelength (hundreds of THz), which is a fundamental manufacturability advantage over coherent systems. Achieving the required miniaturization, however, is both a challenge and an opportunity to drive advancements in 3D optical technologies with broader applications.”

On cost implications, we have deliberately chosen to build AOC using consumer-grade optical and electronic components with mature wafer-scale manufacturing processes. The existing manufacturing ecosystems and volumes for these components indicates a promising path for low cost and mass production. Additionally, in the early stages of hardware innovation, energy consumption estimates often serve as a reasonable proxy for cost, and they eventually factor into operational expenditures (OPEX) – a significant cost for large AI clusters. Consequently, the expected cost reduction could potentially follow a similar trend to the projected energy efficiency improvements. The updated Discussion section says:

“All AOC components including microLEDs, photodetectors, SLMs, and analog electronics, have an existing and growing manufacturing ecosystem with wafer-scale production. At the same time, complementing optics with analog electronics offers numerous opportunities to expand the compute primitives, including nonlinearities, the hardware can support, thereby enhancing its expressiveness.”

[27] A discussion on hardware limitations, error tolerance, and trade-offs in precision would provide a more comprehensive perspective on AOC’s practical deployment.

Hardware limitations, and trade-offs in precision: It is indeed important that the manuscript reflects the main limitations of AOC, which are as follows:

- AOC is not a general-purpose computer: it currently supports a limited set of mathematical operations (e.g. matrix-vector multiplication, tanh nonlinearity, annealing, etc).

- ML and optimization problems need to be formulated within AOC's abstraction (fixed-points models and QUMO, respectively) and within 9 bit precision (including the sign).

These limitations reflect the fact that we have specialized our hardware to target two workload classes. In return, AOC could offer significant speed and energy efficiency improvements. AOC's design thus exploits the trade-off between generality and increased efficiency that is fundamental in hardware design and pushes it further in the context of analog hardware. We agree with the Reviewer that clarifying this will help and looking ahead, expanding the expressiveness of the current hardware is important. In future work, we plan to enlarge the portfolio of operations of AOC to include normalization, softmax, and additional functions that are relevant for optimisation and ML workloads and can be implemented efficiently in the analog domain.

Thanks to the Reviewer's feedback, we have added this the following to the Introductory text and the discussion in Section A.1 Scaling, compute and energy estimations in Supplementary, respectively:

“
By eliminating digital-analog conversions and merging compute and memory to bypass the von Neumann bottleneck, AOC can achieve substantial efficiency gains albeit specialized.
”

“We note that these potential speed and efficiency gains come from exploiting the trade-off between generality and efficiency that is fundamental in hardware design. AOC explores this trade-off by specializing the hardware for the specific tasks of optimization and machine learning and pushing it further in the context of opto-analog hardware.
”

Further, we have added the following in Section B.2 Analog electronic subsystem for nonlinear, annealing, summing, and differencing operations:

“In future work, we plan to expand the operation capabilities of this module to include normalization, softmax, and additional functions that are relevant for optimisation and ML workloads and can be implemented efficiently in the analog domain.”

Error tolerance: Noise and component variation in optical and electrical components pose significant challenges in the development of analog hardware. As discussed in the manuscript, we believe that co-designing the hardware alongside the models it implements is crucial to addressing these challenges. To this end, we developed fixed-point search models for both machine learning and optimization applications. These models exhibit an inherently attractive nature, enhancing tolerance against noise and non-idealities. We have elevated the figure showing these tolerance results to Extended Data Figure 6.

Variation in each hardware component, commonly referred to as component tolerance, is another crucial factor in analog hardware design. As the system is built, these variations must be carefully characterized and calibrated to ensure proper compensation.

Following the Reviewer's feedback, we have added this discussion to Section Evaluation of matrix-vector multiplication accuracy in Method:

“We characterized and calibrated the key opto-electronic and electronic components to equalize the gain of each AOC path. For example, we calibrate the optical paths by applying a set of 93 reference matrices and for each we digitally compute the result of the vector-matrix product. We then adjust the gain per channel slightly so that, averaged over the set of 93 computed vectors, the AOC result is as close as possible to the digital result.

Following this, the accuracy of the matrix-vector multiplication is characterized using the same 93 reference matrices on each SLM and measuring the output of the system, shown in Supp. Fig. 3a.”